# De novo variants in *LRRC8C* resulting in constitutive channel activation cause a human multisystem disorder

Mathieu Quinodoz [1,2,3,11], Sonja Rutz[4,11], Virginie Peter [1,2,11], Livia Garavelli[5,11], A Micheil Innes [6,11], Elena F Lehmann [4], Stephan Kellenberger [7], Zhong Peng[7], Angelica Barone[8], Belinda Campos-Xavier [9], Sheila Unger[9,10], Carlo Rivolta [1,2,3✉], Raimund Dutzler [4✉] & Andrea Superti-Furga [9,10✉]

## Abstract

**Volume-regulated anion channels (VRACs) are multimeric proteins composed of different paralogs of the LRRC8 family. They are activated in response to hypotonic swelling, but little is known about their specific functions. We studied two human individuals with the same congenital syndrome affecting blood vessels, brain, eyes, and bones. The LRRC8C gene harbored de novo variants in both patients, located in a region of the gene encoding the boundary between the pore and a cytoplasmic domain, which is depleted of sequence variations in control subjects. When studied by cryo-EM, both LRRC8C mutant proteins assembled as their wild-type counterparts, but showed increased flexibility, suggesting a destabilization of subunit interactions. When co-expressed with the obligatory LRRC8A subunit, the mutants exhibited enhanced activation, resulting in channel activity even at isotonic conditions in which wild-type channels are closed. We conclude that structural perturbations of LRRC8C impair channel gating and constitute the mechanistic basis of the dominant gain-of-function effect of these pathogenic variants. The pleiotropic phenotype of this novel clinical entity associated with monoallelic LRRC8C variants indicates the fundamental roles of VRACs in different tissues and organs.**

**Keywords** Volume-regulated Anion Channels; Disease-causing Variants; Channel Activation
**Subject Categories** Genetics, Gene Therapy & Genetic Disease; Structural Biology

## Introduction

Volume-regulated anion channels (VRACs) constitute a diverse family of ion channels that are ubiquitously expressed in mammalian cells. They are believed to be essential for maintaining cellular volume (Hoffmann et al, 2009), as they are known to open in response to hypotonic stress, enabling efflux of anions and osmolytes, thereby preventing excessive cell swelling and burst (Jentsch, 2016; Osei-Owusu et al, 2018; Pedersen et al, 2016; Strange et al, 1996). VRACs are hexamers composed of five closely related paralogues of the leucine-rich repeat-containing protein 8 family (termed LRRC8A to E) (Qiu et al, 2014; Voss et al, 2014). In a cellular context, these proteins form heteromers, all containing the obligatory A subunit and at least one other subunit, to obtain functional channels (Planells-Cases et al, 2015; Voss et al, 2014). The general architecture of these ion channels has been defined in the structures of homomers of LRRC8A, which assembles as a hexamer of tightly interacting subunits with modular organization consisting of a membrane-inserted pore domain (PD) and cytoplasmic leucine-rich repeat domains (LRRDs) (Deneka et al, 2018; Kasuya et al, 2018; Kefauver et al, 2018; Kern et al, 2019). Homomeric LRRC8A-only channels show reduced activation properties and respond poorly to cell swelling, suggesting that they maintain a stable closed conformation (Deneka et al, 2018; Syeda et al, 2016; Yamada et al, 2021). In contrast, when expressed on its own, the LRRC8C subunit forms a larger, heptameric assembly with weaker mutual subunit interactions and a different preferential arrangement of the LRRDs (Rutz et al, 2023; Takahashi et al, 2023). Structural studies in heteromeric channels, which contain one or two C-subunits in a hexameric assembly, showed that A- and C-subunits can interact and that the closed conformation of A subunits can be loosened by interaction with the C-subunit (Kern et al, 2023; Rutz et al, 2023). In essence, the

[1]Institute of Molecular and Clinical Ophthalmology Basel (IOB), 4031 Basel, Switzerland. [2]Department of Ophthalmology, University of Basel, 4031 Basel, Switzerland. [3]Department of Genetics and Genome Biology, University of Leicester, Leicester LE1 7RH, UK. [4]Department of Biochemistry University of Zurich, 8057 Zurich, Switzerland. [5]Clinical Genetics Unit, Azienda USL-IRCCS of Reggio Emilia, 42123 Reggio Emilia, Italy. [6]Department of Medical Genetics and Pediatrics and Alberta Children's Hospital Research Institute, Cumming School of Medicine, University of Calgary, Calgary, AB T3B 6A8, Canada. [7]Department of biomedical Sciences, University of, Lausanne, 1011 Lausanne, Switzerland. [8]Pediatric Onco-Hematology Unit, Children's Hospital, Parma University Hospital, Parma, Italy. [9]Division of Genetic Medicine, Lausanne University Hospital (CHUV), and University of Lausanne, 1011 Lausanne, Switzerland. [10]Genetica AG, Zurich and Lausanne, Switzerland. [11]These authors contributed equally: Mathieu Quinodoz, Sonja Rutz, Virginie Peter, Livia Garavelli, A Micheil Innes. ✉E-mail: carlo.rivolta@iob.ch; dutzler@bioc.uzh.ch; asuperti@unil.ch

C-subunits appear to reduce the intrinsic stability of the A subunits, and thus facilitate activation of the channels (Rutz et al, 2023). It must be noted that, despite our understanding of the protein structure, the diversity of channel arrangements with distinct subunit compositions, as well as their distribution and their roles in different cell types, are still undetermined. The landscape of variations of the *LRRC8C* gene shows several truncating variants at the heterozygous state in presumedly healthy individuals (Karczewski et al, 2020), albeit their distribution along the gene is not homogeneous, and there are so far no clinical phenotypes reliably associated with LRRC8C variants. As a consequence, the physiologic roles of LRRC8C and VRACs in general have remained enigmatic.

Here we describe two unrelated individuals who have the same undescribed congenital pleiotropic disorder, and who carry unique monoallelic variants of the *LRRC8C* gene. We found that both mutant proteins assemble with LRRC8A to form functional channels. However, such multimers show strongly increased mobility at the structural level, and increased constitutional activation properties at the functional level. The consequent pronounced gain-of-function, leading to channel activity even in isotonic conditions (where wild type (WT) channels are normally closed), likely underlies the disease phenotype.

## Results

### Identification of patients with genetic variants in *LRRC8C*

In the framework of a large-scale, exome-based genetic analysis of patients with rare undiagnosed disorders, we identified a 1-bp heterozygous insertion in the gene *LRRC8C* in an Italian boy (P1) presenting with marked skin telangiectasia and gastrointestinal vascular dysplasia, microcephaly, intellectual disability, skeletal dysplasia with fragile bones, and eye abnormalities including optic atrophy (Fig. 1; Table 1). This DNA variant, NM_032270.4: c.1197dup, p.(Leu400IlefsTer8) (NC_000001.11:g.89713767dup [hg38], NC_000001.10:g.90179326dup [hg19]), produced a shift of the reading frame leading to the change of Leu 400 to Ile and the insertion of six amino acids followed by a premature termination codon (Fig. 2A). Like most variant-induced interruptions of the coding frame located in the last exon of a gene, c.1197dup (from now on, 'LRRC8C^trunc') is predicted to escape nonsense-mediated mRNA decay and result in the production of a truncated protein product (Holbrook et al, 2004) (Fig. 2B–D). Sequencing parental DNA revealed that this microinsertion was a de novo event. In a parallel study, a girl from Canada (P2) presenting with virtually identical clinical features (marked skin telangiectasia, microcephaly and intellectual disability, visual impairment, bone changes, and short stature) was identified (Sobreira et al, 2015) (Fig. 1; Table 1). Notably, the two patients also had a strong facial resemblance to each other (Fig. 1B,C). The female patient carried a different monoallelic variant in LRRC8C, NM_032270.4:c.1168 G>C, p.(Val390Leu) (NC_000001.11:g.89713738 G>C [hg38], NC_000001.10:g.90179297 G>C [hg19]), or LRRC8C^V390L, which also occurred in a de novo manner (Fig. 1A,C,E). Both variants affected positions in the LRRC8C protein that are conserved across all 100 vertebrates from the Multiz alignment (Blanchette et al, 2004) and in the LRRC8A, B, D and E paralogs (Fig. 2C).

Importantly, both LRRC8C^trunc and LRRC8C^V390L are absent from gnomAD, a DNA variation database of more than 750,000 normal controls (Karczewski et al, 2020; Wiel et al, 2019), and they are located in a region of the protein that is particularly intolerant to both loss-of-function and missense variants, as it is characterized by a rate of missense changes that is ~eightfold lower than the rest of the gene, and by the complete absence of frameshift or nonsense variants (Fig. 2A,D,E). A detailed clinical description of the two individuals is provided as a supplementary document (Appendix Clinical Reports). In accordance with the dyadic naming system (Biesecker et al, 2021; Unger et al, 2023), we propose to call this disorder *LRRC8C-related TIMES syndrome* for Telangiectasia, Intellectual disability, Microcephaly and metaphyseal dysplasia, Eye abnormalities, and Short stature.

### Structural properties of LRRC8C disease mutants

The genetic data suggested a possible link between a severe disease phenotype and two adjacent variants of *LRRC8C* in two unrelated, sporadic patients. Although we considered the possibility that the link was spurious (since no disease had been linked to LRRC8C thus far, and the variants were different in nature), we decided to investigate the structural and functional properties of the resulting channels in detail. Although located in the same region of the gene, the two variants have different predicted effects on the translated product. In P1, the insertion of a base yields a frameshift in the amino acid sequence leading to a truncated protein, whereas the replacement of a valine by a leucine at position 390 leads to a moderate increase of the side chain volume without altering the hydrophobic character of the position (Fig. 2D,E). The position of the variants at the boundary between the PD and the LRRD of the subunit, two parts of the protein that form independent units that can be expressed on their own (Deneka et al, 2018; Rutz et al, 2023) (Fig. 2A,D,E), was intriguing. The first question we addressed was whether constructs of LRRC8C carrying either of the two disease-causing variants would fold and assemble into oligomeric channels. Previous studies of the structure of homomeric LRRC8C revealed its assembly as heptamer (Rutz et al, 2023; Takahashi et al, 2023), which is larger than the hexameric organization of other characterized LRRC8 subunits (Deneka et al, 2018; Kasuya et al, 2018; Kefauver et al, 2018; Kern et al, 2019; Liu et al, 2023) and their heteromeric assemblies (Kern et al, 2023; Rutz et al, 2023). To investigate the biochemical properties of LRRC8C constructs carrying the disease variants, we compared the elution properties of overexpressed and purified WT and mutated human LRRC8C subunits as fusion to green fluorescent protein in HEK293 cells and found LRRC8C^WT and LRRC8C^V390L to elute as monodisperse peaks at the same volume, indicating that the presence of the point variant did not interfere with folding and oligomerization (Fig. 3A). In the case of LRRC8C^trunc, we found a similar behavior as for full-length subunits, with a monodisperse peak eluting at somewhat larger volume, which is consistent with the smaller size of a channel that lacks its LRRDs (Figs. 2D and 3A). Together, our data suggest proper folding and assembly of both mutants into the same heptameric state as WT. After confirming their biochemical integrity, we proceeded with the structural characterization of homomeric assemblies of both mutants. Although such homomeric channels do not presumably exist in a cellular environment, we reasoned that they would be better suited to detect the

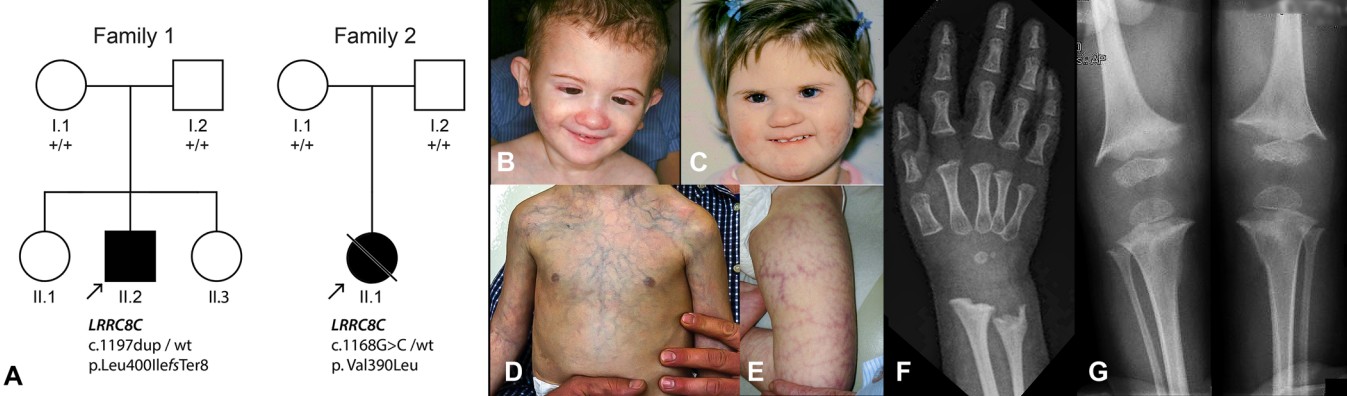

**Figure 1. Genetic and clinical features of the two cases with *LRRC8C* variants.**

(A) Pedigree of patient P1 (Family 1, II.2), of Italian descent, and of patient P2 (Family 2, II.1), a Canadian individual whose parents are from Chile and Canada. The monoallelic de novo *LRRC8C* variants found in the patients are indicated below. (B, C) Facial appearance of P1 (at age 2 year) (B) and P2 (at age 2 years and 10 months) (C) showing short palpebral fissures, a small nose, and smooth facial features. Note the resemblance between the two unrelated individuals. (D, E) Prominent telangiectasia on the trunk (P1) and thigh (P2). (F, G) Radiographs of P1 showing thin diaphyses of the radius and ulna (F) and of the tibia and fibula (G) as well as marked metaphyseal fraying at the distal radius and ulna, distal femur and proximal tibia compatible with metaphyseal dysplasia. See statement in the disclosure and competing interest section at the end of the manuscript for parental consent to publishing these images. Source data are available online for this figure.

consequences of the variants as the LRRC8C subunits in heteromeric channels display a large degree of flexibility (Rutz et al, 2023), which would prohibit a structural interpretation. In independent datasets of each mutant, we found an equivalent heptameric organization as for WT, but also distinct differences in the observed structural features (Fig. 3B–E; Appendix Figs. S1–S3; Table EV1). In both cases, this is manifested in weak density for large parts of the protein, which likely reflects an increased molecular mobility as a consequence of altered subunit interactions (Appendix Figs. S1 and S2). The differences are particularly evident for LRRC8C$^{trunc}$, where the extracellular part of the pore domain carrying the narrow selectivity filter is well-defined as structural unit obeying C7 symmetry (Fig. 3B,C; Appendix Fig. S1). Whereas the extracellular sub-domain (ESD) closely resembles the equivalent region of WT, parts of the protein located further in intracellular direction are not resolved in the density as a consequence of their structural heterogeneity (Fig. 3B,C; Appendix Fig. S1). Even though less pronounced than in LRRC8C$^{trunc}$, related features in the map of LRRC8C$^{V390L}$ indicate a similarly increased subunit mobility (Fig. 3D; Appendix Fig. S2). Akin to the truncated construct, also in case of the missense change, the ESD region is the best-defined part of the structure, obeying C7 symmetry and closely resembling WT. Differences to WT become evident upon comparison of maps of LRRC8C$^{V390L}$ to wild-type LRRC8C homomers processed without application of symmetry (Fig. EV1; Appendix Fig. S3). In the case of LRRC8C$^{WT}$, the entire PDs are well-resolved for six subunits whereas density below the TMD towards the intracellular side is absent in one case (Figs. 3E and EV1A). In this map, five of the LRRDs are defined at medium to high, and one at a lower contour of the map (Figs. 3E and EV1A,B). In contrast, in the map of the best-resolved class of the dataset of LRRC8C$^{V390L}$, the transmembrane regions intracellular to the ESD of three of the seven subunits are not defined, reflecting properties observed in the LRRC8C$^{trunc}$ dataset (Figs. 3D and EV1A). Of the LRRDs, three domains show strong density at medium to high, and

one at a lower contour of the map (Figs. 3D and EV1A,B). These differences illustrate the unexpectedly strong influence of a conservative substitution (valine to leucine) of a residue located at the boundary between both domains of LRRC8C on the conformational properties of the cytoplasmic units, which is presumably coupled to the PD to increase its flexibility (Figs. 2E and 3D,E). Collectively, the data revealed how two very different variants have similar effects on the structural properties of LRRC8C, which would presumably lead to comparable consequences in heteromeric channels containing these subunits.

## Impact of disease-associated variants on LRRC8A/C channel function

After characterizing the structural properties of both disease-causing variants, we investigated whether they would affect the assembly with the obligatory subunit LRRC8A into heteromeric channels and whether these would show altered functional properties. To this end, we studied the formation of LRRC8A/C heteromers by pull-down experiments where we overexpressed channels in HEK293 cells by co-transfection of DNA coding for constructs of either subunit, followed by affinity purification of LRRC8C and detection of the co-purified LRRC8A (Fig. 4A,B). The results shown in Fig. 4B demonstrate the presence of both subunits in the pulled-down samples thus emphasizing that neither variant prevented LRRC8C from forming heteromeric complexes. This is also illustrated in size exclusion chromatography experiments where heteromeric channels containing the mutated subunits eluted with similar properties as WT at a volume corresponding to a hexameric organization (Fig. 4C).

Since neither variant perturbed the assembly of the protein, we investigated the functional properties of LRRC8A/C channels carrying the disease mutants in LRRC8$^{-/-}$ cells (Voss et al, 2014) overexpressing both subunits by patch-clamp electrophysiology in the whole-cell configuration (Figs. EV2 and EV3). In these experiments,

**Table 1. Clinical features.**

| | Patient 1 | Patient 2 |
|---|---|---|
| Sex | Male | Female |
| Genetic variant | LRRC8C$^{trunc}$ | LRRC8C$^{V390L}$ |
| **Clinical findings** | | |
| Telangiectasia | Skin (with cutis marmorata; Fig. 1) and gastrointestinal (GI) tract (duodenal, ileal and colic microvascular ectasia) | Skin (with cutis marmorata, Fig. 1); improvement after introduction of amlodipine at age 15 years |
| Intellectual disability | Moderate intellectual disability; walking acquired but motor skills hampered by visual impairment. Language acquisition delayed, can speak in sentences. | Marked intellectual disability; spastic movement disorder; seizures in infancy and childhood; eventually was able to walk and ride her modified bicycle. Pleasant personality, could make short sentences. |
| Microcephaly | At 12 years 2 months, head circumference 48.5 cm ( <1st percentile, −3.6 SD). | Severe |
| Metaphyseal dysplasia | Metaphyseal dysplasia with gracile diaphyses | Metaphyseal dysplasia with gracile diaphyses |
| Eye findings | At age 10 months, optic nerve atrophy and inferior entropion (operated). At age 12 years, severe bilateral visual impairment, high myopia with astigmatism, bilateral keratoconus, corneal changes with epithelial edema and paracentral corneal leucoma, pigmentary alterations of the fundus, and glaucoma (operated). At age 16 years corneal transplant on the right eye. | Optic atrophy; deep-set eyes, hypotelorism |
| Short stature | At 12 years, height 112 cm ( <1st percentile, −5.1 SD). | (< 3rd percentile) |
| Facial features | Short, upturned nose | Short, upturned nose; upslanting palpebral fissures; cupped ears |
| Other | - Congenital anal atresia (operated)<br>- Recurrent GI bleeding with severe anemia (Hb up to 2.1 g/dl)<br>- Toe syndactylies (II–III and IV–V on the right side and II–III on the left side)<br>- Osteopenia with multiple fractures<br>- Hypothyroidism<br>- EEG: temporo-parieto-occipital abnormalities | - Severe arterial hypertension since age 15<br>- Recurrent urinary tract infections and recurrent renal calculi<br>- EBV infection with autoimmune liver disease<br>- Died at age 20 years with Influenza A-related respiratory insufficiency and gastric necrosis |

we recorded VRAC currents that were evoked in response to the perfusion of cells with hypotonic buffers of two different compositions, both of which were previously used to induce swelling and the consequent slow activation of endogenous channels (Sukalskaia et al, 2021; Voss et al, 2014) (Fig. EV3A,B). In case of hLRRC8A/C$^{WT}$, this protocol resulted in a robust and reversible swelling-activated current response in one set of solutions with reduced salt concentration in the hypotonic condition (Δsalt), showing the characteristic properties of LRRC8A/C channels with pronounced outward rectification and weak inactivation at positive voltages (Voss et al, 2014) (Figs. 4D and EV2A). No response of hLRRC8A/C$^{WT}$ channels was detected in the second set of solutions with reduced mannitol concentration in the hypotonic condition (Δmannitol) (Fig. EV3C,D), which differs from the robust currents of murine orthologs obtained in equivalent conditions in a previous study (Sukalskaia et al, 2021). When using the same protocol for LRRC8A/C heteromers containing either one of the two disease-related constructs of LRRC8C, we find a consistently higher current density in both buffer compositions, with pronounced activity already at isotonic conditions that is further increased much beyond WT in swollen cells (Figs. 4E–H, EV2B–E, and EV3E–H). The basal currents share the same IV relationships and anion selectivity properties as currents of WT and mutated constructs obtained after exposure to hypotonic conditions (Fig. EV2A–F). In addition, their inhibition by DCPIB provides further evidence that they originate from open VRAC channels (Fig. EV2G).

Since the gnomAD database contains numerous nonsense variants of LRRC8C resulting in truncated proteins of different length that do not cause detectable symptoms, we have investigated

the biochemical and functional properties of two variants that are closest to LRRC8C$^{trunc}$, one leading to a construct that is 21 residues shorter (termed LRRC8C$^{t379}$) and another that is 19 residues longer (termed LRRC8C$^{t419}$) (Fig. EV4A). We first investigated the biochemical properties of both constructs by monitoring their elution on size exclusion chromatography after overexpression and solubilization. In these experiments, we found robust assembly similar to LRRC8C$^{trunc}$ in case of the longer protein construct LRRC8C$^{t419}$, whereas no protein was detected in case of LRRC8C$^{t379}$, indicating a compromised expression and folding of the shorter protein construct (Fig. EV4B). When investigating heteromeric channels containing either construct by patch-clamp electrophysiology in buffer system Δsalt, we found in both cases a consistent behavior with very low currents under isotonic conditions that are barely above WT and much smaller than in LRRC8C$^{trunc}$ and little further activation upon change to hypotonic conditions, suggesting that both mutations resulted in functionally compromised channels that do not share the strong gain-of-function exerted by the LRRC8$^{trunc}$ or LRRC8$^{V390L}$ variants (Fig. EV4C).

Together, our results on the functional characterization of heteromeric LRRC8A/C channels demonstrate a similar activating phenotype for both disease-associated variants upon their overexpression in a cell line that does not contain endogenous subunits. The fact that this phenotype is not shared by constructs that are moderately smaller or larger than LRRC8C$^{trunc}$, points towards the importance of the mutated region as hotspot for channel activation. This strong gain-of-function is presumably also present in a complex physiological context, where cells express different

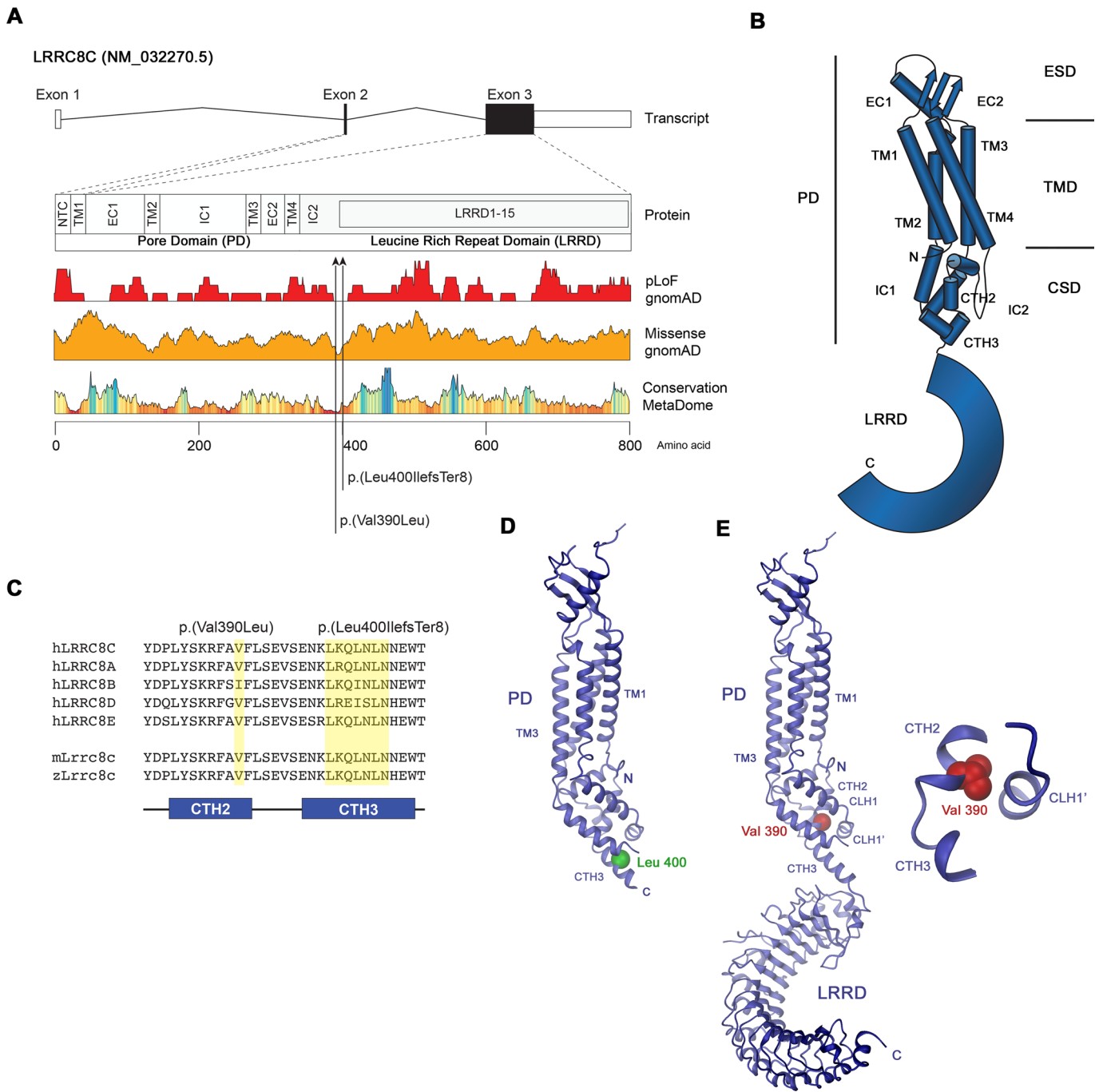

**Figure 2. Characteristics of LRRC8C and of the variants identified.**

(A) Schematic representation of the main *LRRC8C* transcript and its protein product. Both LRRC8C^trunc (p.Leu400IlefssTer8) and LRRC8C^V390L (p.Val390Leu), depicted by black arrowheads, fall into a region of the protein that is bereft of loss-of-function (pLoF, red track) or missense (orange track) variants in normal controls, as reported in the gnomAD database of normal controls, likely representing a sensitive site for DNA changes to result in pathogenic variants, as also shown by the MetaDome track (red, highest intolerance to changes; blue, lowest intolerance to changes). NTC N-terminal coil, TM transmembrane domain, EC extracellular loop, IC intracellular loop, LRRD leucine-rich repeat domain. (B) LRRC8C topology with secondary structure element of the pore PD indicated and the LRRD approximated as blue arc. (C) Conservation of the CTH2 and CTH3 region domains, with respect to the amino acid residues impacted by LRRC8C^trunc and LRRC8C^V390L, in human paralogues of LRRC8C (hLRRC8x) and orthologues from mouse (mLrrc8c) and zebrafish (zLrrc8c). LRRC8C is not present outside of bony vertebrates. (D) Model of the protein construct LRRC8C^trunc based on the murine LRRC8C^WT structure (PDBID: 8B40). The position of Leu 400 is indicated as a green sphere. (E) Structure of the murine LRRC8C^WT subunit with the position of Val 390 indicated as red sphere. Inset (right) shows blowup of the region around Val 390 with its side chain represented as CPK model.

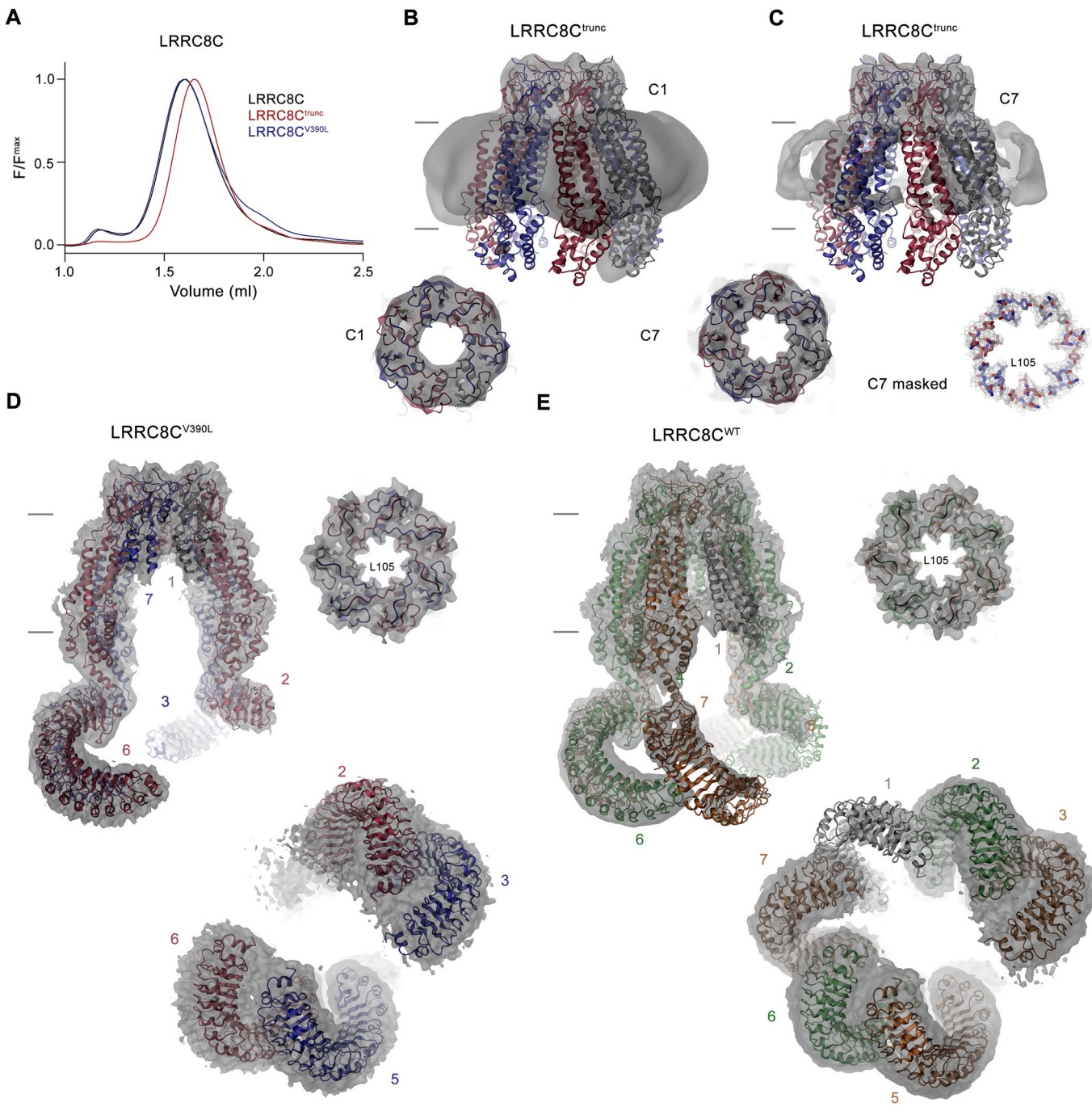

**Figure 3.   Structural properties of LRRC8C disease mutants.**

(**A**) Size exclusion profiles of detergent-solubilized constructs of human LRRC8C$^{WT}$ and the two disease constructs LRRC8C$^{trunc}$ and LRRC8C$^{V390L}$, each containing a C-terminal fusion to GFP. The elution of extracted proteins was detected via their GFP-fluorescence. (**B**, **C**) Structure of a homomeric hLRRC8C$^{trunc}$ construct. The cryo-EM densities at 7.3 Å prior to the application of symmetry (**B**) and at 4.9 Å after the application of C7 symmetry (**C**) are shown superimposed on a model of the PD of the heptameric protein viewed from within the membrane (top) and from the extracellular side (bottom, left). The density of the filter at 3.4 Å, obtained after masking, is shown on the bottom, right (**C**). Although the ESD is well-defined, the largest parts of the TM and the CSD are absent. (**D**) Cryo-EM density of a homomeric structure of hLRRC8C$^{V390L}$ at 4.4 Å processed without application of symmetry (C1). (**E**) Cryo-EM density of hLRRC8C$^{WT}$ at 3.7 Å processed without application of symmetry (C1). (**D**, **E**) Left, view from within the membrane contoured at 6.5 σ. Right, view from the extracellular side at 8.5 σ with model shown as ribbon and sidechains of the constricting residues Leu 105 as sticks. Bottom, right, view of the LRRDs from the cytoplasm at lower contour (3.5 σ). The relative positions of subunits within the heptamer are indicated. Source data are available online for this figure.

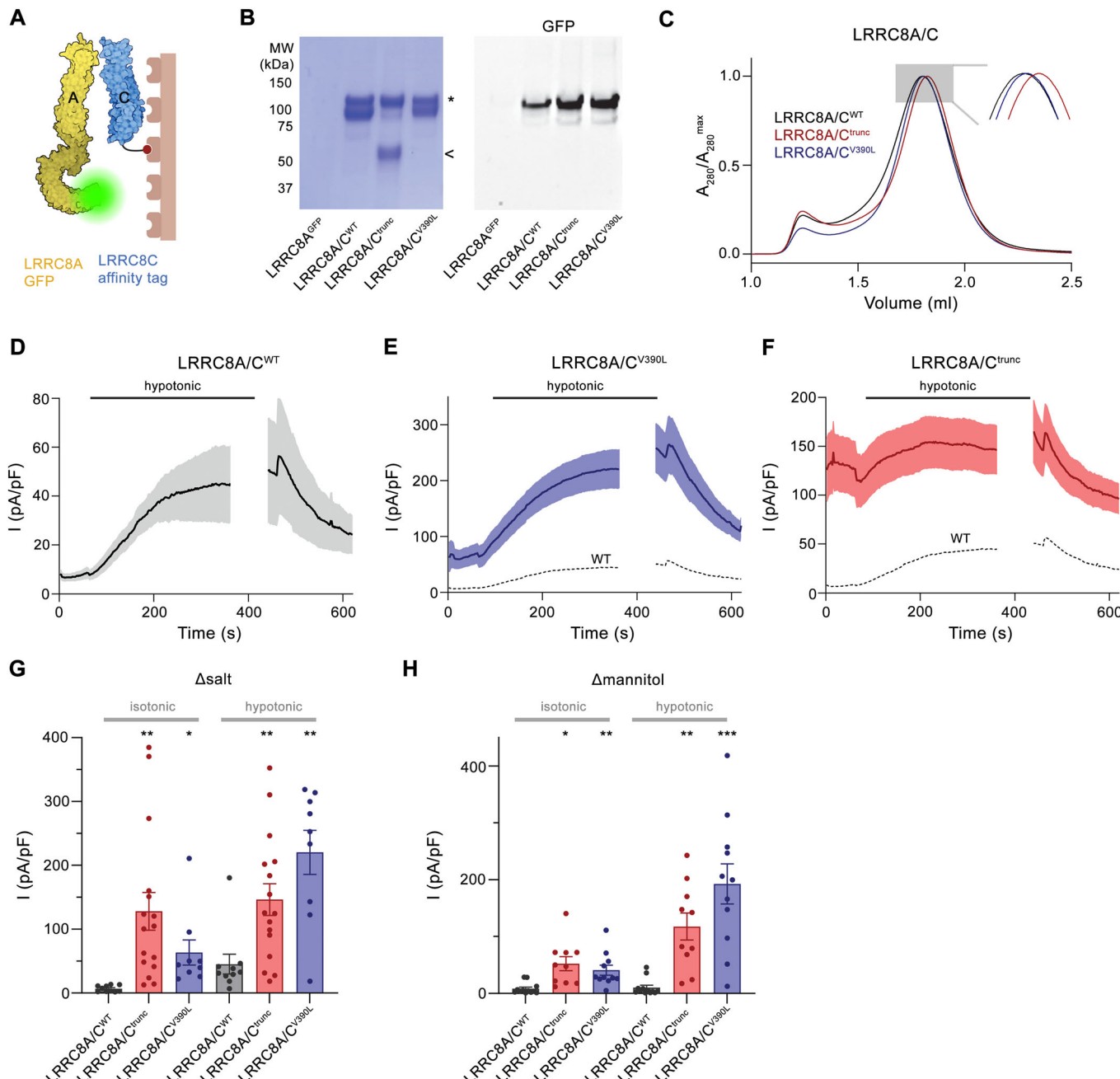

LRRC8 subunits, which provides a plausible explanation of the similar disease phenotype caused by the two different variants in heterozygosity and affecting a subunit hitherto considered as non-essential.

## Discussion

Our work provides evidence linking a novel syndromic condition, for which we propose the name of *LRRC8C*-related TIMES syndrome (an acronym for Telangiectasia, Intellectual disability, Microcephaly and metaphyseal dysplasia, Eye findings and Short

stature), with variants in a volume-regulated anion channel subunit that is a priori involved in basic and ubiquitous functions related to the regulation of the cellular size in homeostatic conditions (Jentsch, 2016). At the molecular level, the identified variants result in a classic gain-of-function effect, with a potential dominant mechanism of heredity following de novo events. This specific genetic mechanism is typical of severe conditions in which reproductive fitness is strongly reduced or absent. Haploinsufficiency, i.e., pathogenicity due to variant-induced insufficient protein production, can be excluded as two control individuals carrying loss-of-function variants in exon 2 (the first coding exon) have been reported in the gnomAD database (v.2.1.1) (Karczewski

◄

**Figure 4. Heteromeric assembly and function of LRRC8C disease variants.**

(A) Strategy for the investigation of heteromerization of LRRC8C disease variants. LRRC8C constructs carry an SBP tag for affinity purification and LRRC8A contains a fusion to GFP for detection. (B) Results from a single-step affinity purification. Left, Coomassie-stained SDS-PAGE gel showing both subunits. The LRRC8A-GFP and full-length LRRC8C constructs run at a similar molecular weight around 100 kDa (*), the LRRC8C$^{trunc}$ construct (<) at 50 kDa. Right, detection of in-gel fluorescence of GFP-tagged LRRC8A subunits demonstrates its pulldown by different LRRC8C constructs with similar efficiency. LRRC8A$^{GFP}$ refers to cells that were only transfected with LRRC8A-GFP as negative control. (C) SEC chromatogram of purified heteromeric LRRC8A/C channels containing WT and disease constructs of LRRC8C. Inset shows blowup of the peak region. (D–F) Current densities (at 100 mV) of LRRC8$^{−/−}$ cells transfected with constructs coding for hLRRC8A and either hLRRC8C$^{WT}$ or disease mutants assayed by patch-clamp electrophysiology in the whole-cell configuration. Currents were periodically recorded by a ramp protocol. Perfusion with hypotonic medium is indicated by bar. Traces show mean and s.e.m. of (D), LRRC8A/C$^{WT}$ ($n = 10$), (E), LRRC8A/C$^{V390L}$ ($n = 10$), (F), LRRC8A/C$^{trunc}$ ($n = 16$) biological replicates recorded in buffer system 1 (Δsalt, 50 mM/kg gradient). (E, F) Mean current density of WT is shown as dashed line for comparison. (G, H) Mean current density (at 100 mV) in isotonic (measured 10 s after establishment of whole-cell configuration) and hypotonic conditions (measured 4 min after switch to hypotonic solutions) in (G), buffer system 1 (Δsalt) or (H), buffer system 2 (Δmannitol, 80 mM/kg gradient). Panels show measurements of individual cells (spheres), bars represent mean currents, errors are s.e.m. Asterisks indicate significant deviations from WT in the respective conditions in a Brown–Forsythe and Welch's ANOVA test with Dunnett's T3 multiple comparison testing *$P < 0.05$; **$P < 0.01$; ***$P < 0.001$ (Δsalt isotonic: LRRC8C$^{trunc}$ $P = 0.0019$, LRRC8C$^{V390L}$ $P = 0.0406$, Δsalt hypotonic: LRRC8C$^{trunc}$ $P = 0.0041$, LRRC8C$^{V390L}$ $P = 0.0014$, Δmannitol isotonic: LRRC8C$^{trunc}$ $P = 0.012$, LRRC8C$^{V390L}$ $P = 0.008$, Δmannitol hypotonic: LRRC8C$^{trunc}$ $P = 0.0025$, LRRC8C$^{V390L}$ $P = 0.0009$). The current increase upon exposure to hypotonic conditions in buffer system Δsalt was significant for LRRC8A/C$^{WT}$ ($P = 0.035$) and LRRC8A/C$^{V390L}$ ($P = 0.0017$) but not in case of LRRC8A/C$^{trunc}$ ($P = 0.639$). In buffer system Δmannitol, significant changes were observed for LRRC8A/C$^{V390L}$ ($P = 0.0015$) and LRRC8A/C$^{trunc}$ ($P = 0.0296$) but not LRRC8A/C$^{WT}$ ($P = 0.721$). Source data are available online for this figure.

et al, 2020). A dominant-negative effect can also be ruled out because the variants identified did not abolish the natural LRRC8C functions but, conversely, enhanced them.

It is interesting to note that as both LRRC8C$^{trunc}$ and LRRC8C$^{V390L}$ allow the multimerization of the VRAC complex (Figs. 3 and 4B,C), the electrophysiological phenotype at a cellular level and the clinical phenotype are probably a direct consequence of the altered functional properties resulting from the abnormal composition of the channel (Fig. 4D–H). In particular, the disease mechanism is dependent on the preservation of the entire (and functional) pore domain of LRRC8C while the cytoplasmic LRRD domain is lost (as in LRRC8C$^{trunc}$) or a variant at the boundary between the two domains, which might compromise the functional coupling between both units (as in LRRC8C$^{V390L}$). This is corroborated by the presence of healthy individuals carrying heterozygous truncating variants of LRRC8C upstream or downstream of the LRRC8C$^{trunc}$ site (Karczewski et al, 2020) (Figs. 2A and 5A). For variants located in the PD, it is expected that the resulting protein would not be able to fold and multimerize and thus would be non-functional. For variants located downstream in the LRRD, it is unclear whether the presence of these variants in normal controls is due to some residual function of the subunits that is retained, or to the inability of these mutant versions of LRRC8C to multimerize and therefore interfere with the physiological functions of VRAC; these would be then true null alleles. In case of severely compromised folding, these variants could lead to degradation. Clearly, identifying the LRRC8C$^{trunc}$ frameshift in a region that is bereft of any other types of DNA variants in the general population implies that this specific region is crucial to VRAC's normal functions. In this respect, a previously described spontaneous mutation of LRRC8A leading to a premature stop removing the bulk of the LRRD in *ébouriffé* mice is relevant since this construct, which is somewhat longer than the variant LRRC8C$^{t419}$ leads to a similar severe reduction of activity (Fig. EV4A,C). In this case, the low activity was assigned in part to the compromised assembly, which has interfered with the trafficking of LRRC8 channels to the plasma membrane (Luck et al, 2018; Platt et al, 2017). In contrast to LRRC8C$^{trunc}$, this variant is recessive, and no phenotype was observed in heterozygous mice.

Despite the distinct difference between the two genetic variants, one (LRRC8C$^{V390L}$) resulting in a conservative replacement of amino acid, the second (LRRC8C$^{trunc}$) in the loss of an entire protein domain (representing ~50% of the protein), their structural and functional consequences are similar, leading to a strong increase of the subunit mobility in an oligomeric assembly (Fig. 3) and in consequence to a pronounced increase of activity even under isotonic conditions where VRACs are usually closed (Fig. 4). The location of both variants at the boundary between two autonomous folding units further emphasizes the role of the cytoplasmic LRRDs as allosteric regulators of activity (Deneka et al, 2021; Gaitan-Penas et al, 2016; Konig et al, 2019; Sawicka and Dutzler, 2022). The functional importance of the affected region as hotspot for activation is also reflected in the pronounced difference found in variants resulting from missense mutations located immediately up- or downstream of the site mutated in LRRC8C$^{trunc}$, which in both cases leads to a pronounced loss of activity (Fig. EV4C). Our data thus support the role of LRRC8 subunits other than the obligatory LRRC8A in increasing the activation properties of the channels by destabilization of the A subunits, as has been suggested previously (Rutz et al, 2023) (Fig. 5B). Our data also explicitly confirm that the LRRC8C subunit is of critical importance for the control of channel activity.

The ubiquitous expression of VRACs in vertebrate cells has suggested critical roles for the channel, with several studies using knockout animals providing initial insight into specific functions in which VRACs may participate (Alghanem et al, 2021; Balkaya et al, 2023; Chu et al, 2023; Concepcion et al, 2022; Knecht et al, 2024; Lopez-Cayuqueo et al, 2022). However, despite these extensive studies, its role in many physiologic processes it is involved in have remained enigmatic. The clinical features observed in the two patients thus provide a first glimpse of the importance of VRACs in humans as they seem to indicate that these anion channels are crucial in different cell types in development and homeostasis. Educated guess would suggest that diverse cell types such as endothelial cells in capillary vessels, chondrocytes undergoing hypertrophy in the growth plate, and neural progenitor cells contributing to brain growth and development may be dependent on physiologic processes affected by VRAC dysfunction. In theory,

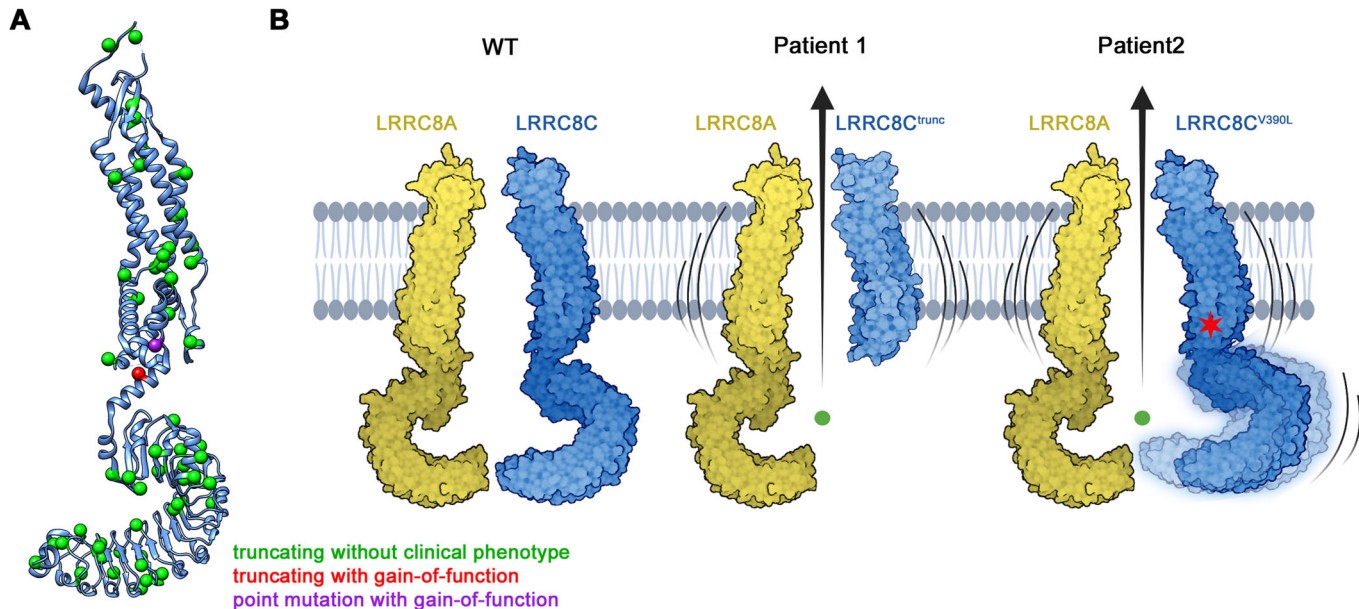

**Figure 5. Phenotype of human LRRC8C variants.**

(A) Distribution of variants identified in human LRRC8C and their phenotypical consequence. Positions of variants are mapped on the LRRC8C subunit and shown as spheres. Variants leading to a premature termination of the protein without clinical phenotype (as ascertained in gnomAD, see text) are colored in green, the two variants discussed in this study with strong gain-of-function phenotype in red and violet. (B) Schematic depiction of the effect of disease variants in LRRC8C with pronounced gain-of-function phenotype on channel activity. Two subunits of a VRAC heteromer are shown, where the two LRRC8C disease mutants increase the flexibility of the subunit, leading to channel opening already under isotonic conditions.

inappropriate activation of the channel might lead to exaggerated loss of water and anions and thus to cell shrinkage. A constitutively open anion channel might also impact the membrane potential of the cell or allow for the uptake or efflux of signaling compounds at inappropriate times (Concepcion et al, 2022; Lutter et al, 2017; Zhou et al, 2020), or perhaps interfere with (or induce) apoptotic processes (Okada et al, 2001; Okada et al, 2004; Poulsen et al, 2010). Which mechanisms ultimately lead to cell dysfunction and subsequent macroscopic defects at the tissue and organ levels remains to be investigated. Such studies might also reveal approaches to pharmacologic modulation of VRAC hyperactivity.

## Methods

### Reagents and tools table

| Reagent/resource | Reference or source | Identifier or catalog number |
| --- | --- | --- |
| **Experimental models** | | |
| **Recombinant DNA** | | |
| Human LRRC8A open reading frame | GenScript | GenID: 56262 |
| Human LRRC8C open reading frame | GenScript | GenID: 84230 |
| Patient 1 | Cloned from human LRRC8C | n.a. |
| Patient 2 | Cloned from human LRRC8C | n.a. |

| Reagent/resource | Reference or source | Identifier or catalog number |
| --- | --- | --- |
| Mammalian expression vector with C-terminal streptavidin-binding peptide, myc tag, and 3 C cleavage site | Dutzler Laboratory | n.a. |
| Mammalian expression vector with C-terminal streptavidin-binding peptide, myc tag, Venus-tag and 3 C cleavage site | Dutzler Laboratory | n.a. |
| Mammalian expression vector with C-terminal 3C cleavage site, GFP-tag, and myc tag | Dutzler Laboratory | n.a. |
| pINIT | Dutzler Laboratory | n.a. |
| **Antibodies** | | |
| **Oligonucleotides and other sequence-based reagents** | | |
| Primers for the generation of patient construct | Microsynth | n.a. |
| **Chemicals, enzymes, and other reagents** | | |
| ATP, disodium salt | AppliChem | A1348.0005 |
| Benzamidine | Sigma | B6506 |
| Calcium chloride | Sigma | 223506 |
| Cesium chloride | Sigma | 289329-100G |
| Cs-Met-sulfonate | Sigma | C1426-25G |
| D-desthiobiotin | Sigma | D1411 |

| Reagent/resource | Reference or source | Identifier or catalog number |
|---|---|---|
| Digitonin High Purity | Merck Millipore | 300410 |
| DNase I | AppliChem | A3778 |
| Dulbecco's modified Eagle's medium - high glucose, glutamine and pyruvate supplemented | Gibco, ThermoFisher Scientific | 41966-029 |
| EGTA | Sigma | 03777-50G |
| Fetal bovine serum | Sigma | F7524-500mL |
| GDN | Anatrace | GDN101 |
| Glucose | AppliChem | A1422,1000 |
| HCl | Merck Millipore | 1.00319.1000 |
| HEPES | Sigma | H3375 |
| HiSeq Rapid PE Cluster Kit v2 | Illumina | |
| HRV 3C protease | Expressed (pET_3C) and purified in Dutzler laboratory | n.a. |
| HyClone HyCell TransFx-H medium | Cytiva | SH30939.02 |
| Kolliphor P188 | Sigma | K4894 |
| Leupeptin | AppliChem | A2183 |
| L-glutamine | Sigma | G7513 |
| Lipofectamine 2000 | Invitrogen | 11668019 |
| Magnesium Chloride | Fluka | 63065 |
| D-Mannitol | Sigma | M1902-500G |
| NMDG | Sigma | 66930-100g |
| OPTI-MEM I (1x) Reduced Serum Medium | Gibco | 31985-062 100 mL |
| PEI Max | Fisher Scientific | NC1038561 |
| Penicillin–streptomycin | Sigma | P0781 |
| Pepstatin | AppliChem | A2205 |
| Phosphate buffer saline, sterile calcium free | Sigma | D8537 |
| PMSF | Sigma | P7626 |
| Polyethyleneimine MAX 40 kDa | PolySciences Inc | 24765–1 |
| Potassium chloride | Sigma | 746346 |
| Sodium chloride | Sigma | 71380 |
| SureSelect Human A11 Exon v6 kit | Agilent | |
| Tris | AppliChem | A1379 |
| Valproic acid | Sigma | P4543 |
| **Software** | | |
| Clampex 10.6 | Molecular Devices | n.a. |
| Clampfit 10.6 / 11.0.3 | Molecular Devices | n.a. |
| GraphPad Prism 10.0.2 | GraphPad | 10.0.2 |
| Excel | Microsoft | 2019 |
| ASTRA7.2 | Wyatt Technology | https://www.wyatt.com/products/software/astra.html; RRID:SCR_016255 |
| BioRender | | BioRender.com |
| Chimera v.1.16 | Pettersen et al, 2004 | https://www.cgl.ucsf.edu/chimera/; RRID:SCR_004097 |
| ChimeraX v.1.1.1 | Pettersen et al, 2021 | https://www.rbvi.ucsf.edu/chimerax/; RRID:SCR_015872 |
| Coot v.0.9.4 | Emsley and Cowtan, 2004 | https://www2.mrc-lmb.cam.ac.uk/personal/pemsley/coot/; RRID:SCR_014222 |
| cryoSPARC v.4.3.0. | Structura Biotechnology Inc | https://cryosparc.com/; RRID:SCR_016501 |
| DINO | | http://www.dino3d.org; RRID:SCR_013497 |
| EPU2.9 | ThermoFisher Scientific | n.a. |
| Relion 4.0. | Zivanov et al, 2018 | https://www3.mrc-lmb.cam.ac.uk/relion/ |
| Phenix | Liebschner et al, 2019 | https://www.phenix-online.org/; RRID:SCR_014224 |
| GeneMatcher | Sobreira et al, 2015 | n.a. |
| SimulConsult | | https://simulconsult.com/resources/measurement.html |
| Novoalign v3.08.00 | Novocraft Technologies | n.a. |
| Picard v2.14.0-SNAPSHOT | Broad Institute | n.a. |
| Genome Analysis Toolkit v3.8. (GATK) | Broad Institute | PMID:25431634 |
| **Other** | | |
| HEK293S GnTI⁻ | ATCC | CRL-3022 |
| HEK293T LRRC8⁻/⁻ | Jentsch Lab | n.a. |
| 4–20% Mini-PROTEAN TGX Precast Protein Gels, 15-well, 15 µl | BioRad Laboratories | 4561096DC |
| Amicon Ultra-4 Centrifugal Filters Ultracel 100 K, 4 ml | Merck Millipore | UFC810024 |
| Amicon Ultra Centrifugal Filters Ultracel 100 K, 15 ml | Merck Millipore | UFC9100 |
| QuantiFoil R1.2/1.3 Au 200 mesh | Electron Microscopy Sciences | Q2100AR1.3 |
| Strep-Tactin Superflow high capacity 50% suspension | IBA LifeSciences | 2-1208-010 |
| Superose 6 10/300 GL | Cytiva | 17517201 |
| Superose 6 Increase 5/150 | Cytiva | 29091597 |

| Reagent/resource | Reference or source | Identifier or catalog number |
|---|---|---|
| Ultrafree MC GV 0.22 μm centrifugal filter | Merck Millipore | UFC30GVNB |
| BioQuantum Energy Filter | Gatan | n.a. |
| K3 Summit Direct Detector | Gatan | n.a. |
| 300 kV Titan Krios G3i | ThermoFisher Scientific | n.a. |
| Viber Fusion FX7 imaging system | Witec | n.a. |
| Vitrobot Mark IV | ThermoFisher Scientific | n.a. |
| Borosilicate Glass Capillaries with Filament | Science Products GmbH | BF-150-86-10HP |
| Micropipette Puller | Sutter | n.a. |
| Microforge | Narishige | n.a. |
| Axopatch 200B | Molecular Devices | n.a. |
| Digidata 1440 | Molecular Devices | n.a. |
| NovaSeq 6000 instrument | Illumina | n.a. |

## Patients and families

This study was performed according to the tenets of the Declaration of Helsinki, following the signature of written informed consent forms (including the use of images) by the patients and their family members. Specifically, the research project on exome sequencing in individuals with rare and undiagnosed diseases was approved by the Ethikkommission Nordwest- und Zentralschweiz (project-ID 2019-01660). In both individuals studied, next-generation sequencing was used in a diagnostic context to confirm (or rule out) a list of differential diagnoses considered by the physicians responsible for the clinical workup and management of cases (see Appendix Clinical Reports). In both cases, bioinformatic analysis was opened to all known genes only after all analyses targeted at known clinically relevant genes ("Mendeliome") had yielded negative results. Gene Matcher, an online matching tool, was instrumental in recruiting the Canadian patient (P2) to the study (Sobreira et al, 2015).

## Percentile calculations

Stature, weight, and head circumference percentiles and SD deviations were calculated online at https://simulconsult.com/resources/measurement.html.

## Next-generation sequencing

### P1

Genomic DNA was extracted from peripheral blood leukocytes of the patient and both parents according to standard procedures. Exome capture and library preparation was performed using the SureSelect Human All Exon v6 kit (Agilent, Santa Clara, USA) and HiSeq Rapid PE Cluster Kit v2 (Illumina, San Diego, USA) with 2 μg genomic DNA. Libraries were sequenced on a NovaSeq 6000 instrument (Illumina). Bioinformatic analyses were performed as

described previously (Peter et al, 2019). Briefly, raw reads were mapped to the human reference genome (hg19/ GRCh37) using the Novoalign software (V3.08.00, Novocraft Technologies). Next, Picard (version 2.14.0-SNAPSHOT) was used to remove duplicate reads and Genome Analysis Toolkit (GATK) (version 3.8) (Van der Auwera et al, 2013) was used to perform base quality score recalibration on both single-nucleotide variants and insertion–deletions. A VCF file with the variants was generated by HaplotypeCaller. Then, DNA variants were filtered based on quality, frequency in various databases and on predicted impact on protein sequence and messenger RNA (mRNA) splicing. Finally, they were annotated according to a specific in-house pipeline and filtered for specific inheritance modes.

### P2

Clinical whole-exome sequencing in a CLIA-accredited laboratory did not result in the identification of any known or candidate variants in previously established disease genes. The LRRC8C variant was noticed to be de novo and reported as a Variant of Unknown Significance (VUS) in a gene with no known disease relationships, which led to its upload to the GeneMatcher database (Sobreira et al, 2015).

## Sanger sequencing

All *LRRC8C* variants identified by NGS were validated by the Sanger method, using standard reagents and conditions.

## Generation of expression constructs

Human wild-type LRRC8A and LRRC8C constructs were obtained from GenScript. For construct generation, wild-type LRRC8C was cloned into a pINIT vector containig SapI cloning sites at both ends of the gene of interest by FX cloning (Geertsma and Dutzler, 2011). For the generation of LRRC8C$^{trunc}$, the entire vector was amplified using a forward primer that contains the sequence of the seven newly introduced amino acids followed by a stop codon (IKAAELK*) and a novel Sap1 recognition site (5′-GCTGAACT-TAAAGCATGAAGAGCGGCCACCGAGGCCG-3′). The reverse primer contained the sequence of the newly introduced amino acids followed by the native LRRC8C sequence (5′-TTTAAGTT-CAGCTGCTTTAATTTTGTTTTCACTGACTTCAGACAGGAA-CACTGCAAATCTCTTGG-3′). The construct was subsequently subcloned into pINIT.

LRRC8C$^{t379}$ and LRRC8C$^{t419}$ were generated by PCR amplification of the respective regions of the LRRC8C$^{WT}$ sequence using primers flanking the 5′ end of the native sequence and the respective site of truncation at the 3′ end. The obtained PCR products were subsequently cloned into pINIT. Both variants were obtained using the same forward primer containing a SapI cloning site followed by the sequence of LRRC8C$^{WT}$ (5′-GCATGAA-GAGCGGCCAC-3′). The reverse primers were designed to truncate constructs at respective positions without including additional residues resulting from the nonsense mutations. Both reverse primers contained the complementary sequence prior to the site of truncation followed by a SapI cleavage site (LRRC8C$^{t379}$: 5′-ATCTATCATATGAAGCATAAAAGCAAAGTC-3′, LRRC8C$^{t419}$: 5′-TAGCTTCTGCCTCAGTTTATCAG-3′). Upon cloning into mammalian expression vectors, the C-terminus of LRRC8C$^{t379}$

and LRRC8C[t419] is extended by a single Ala resulting from FX cloning. To generate the LRRC8C[V390L] variant, the point-mutation was introduced using the QuikChange method (Zheng et al, 2004) (forward primer: 5′-GAGATTTGCACTGTTCCTGTCTGAAGTC AGTGAAAACAAATTAAAGCAG-3′, reverse primer: 5′-AGA-CAGGAACAGTGCAAATCTCTTGGAATAGAGAGGGTCATAC TGATC-3′). All generated constructs were subcloned into various expression vectors using FX cloning and confirmed by sequencing (Microsynth). For the expression and purification of homomeric proteins, constructs were cloned into pcDX vectors containing a C-terminal rhinovirus 3 C protease cleavable linker (3 C cleavage site) followed by Venus (Rekas et al, 2002), a myc tag and a streptavidin-binding peptide (Keefe et al, 2001) (SBP, expression vector pcDXc3VMS). For assembly experiments of hLRRC8C mutants with hLRRC8A, the latter was cloned into a pcDX vector containing a C-terminal 3C cleavage site, GFP (green fluorescent protein) and a myc tag (pcDxc3GM). Different hLRRC8C constructs were inserted into a pcDX vector containing a C-terminal 3 C cleavage site, a myc tag and a SBP tag (pcDXc3MS). For electrophysiology experiments, hLRRC8A, hLRRC8C, hLRRC8C[trunc], hLRRC8C[t379], hLRRC8C[t419] and hLRRC8C[V390L] constructs were cloned into a pcDxc3MS vector either inserting a stop codon immediately after the protein sequence or appending the amino acid sequence LSRGPV before the stop codon (which was inserted as a consequence of the cloning strategy). A Venus-only construct was co-expressed in a pcDxc3MS vector.

## Cell culture

Cell-lines were either obtained from ATCC (HEK293S GnT⁻, CRL-3022) or kindly provided by T. J. Jentsch (Lutter et al, 2017; Voss et al, 2014) (HEK293 LRRC8⁻/⁻ cells with all five LRRC8 genes genetically knocked out) and regularly tested for mycoplasma contamination. Cells were grown at 37 °C and 5% CO₂. For initial purification tests of homomeric hLRRC8C constructs and for electrophysiology measurements, adherent HEK293 LRRC8⁻/⁻ cells were grown in high-glucose DMEM medium (Gibco), supplemented with 10% FBS, 4 mM L-glutamine, 1 mM sodium pyruvate and 100 U/ml penicillin–streptomycin. For large-scale homomeric protein expression and assembly studies, HEK293 LRRC8⁻/⁻ cells and HEK293S GnT⁻ cells were adapted to suspension culture and grown in HyCell Hyclone TransFx-H medium (Cytivia) supplemented with 1% FBS, 4 mM L-glutamine, 100 U/ml penicillin–streptomycin and 1.5 g/l Kolliphor-188.

## Protein overexpression and purification

For the initial purification tests of homomeric hLRRC8C mutants, constructs were transfected into adherent HEK293 LRRC8⁻/⁻ cells at 100% confluency. For each 10-cm dish, a total of 10 µg DNA was mixed with 25 µg PEI MAX in 500 µl of non-supplemented DMEM, incubated for 15 min, and carefully distributed over the cells together with valproic acid (VPA) at a final concentration of 3.3 mM. Cells were harvested two days after transfection and washed with PBS prior to protein extraction.

For large-scale purification of homomeric hLRRC8C variants and for assembly studies, constructs were transfected into suspension HEK293 LRRC8⁻/⁻ cells (for structural analysis) or suspension HEK293S GnT⁻ cells (for assembly studies) at hLRRC8A to hLRRC8C plasmid ratios of 1:2 using a similar approach as described for adherent cells. Cells were split one day before transfection to a density of $5 \times 10^5$ cells/ml. For transfection, polyethyleneimine (PEI MAX, 40 kDa) was mixed with the purified DNA at a ratio of 1:2.5 (w/w) in non-supplemented DMEM at 0.01 mg DNA per ml to obtain a final ratio of 1.2 µg DNA per million of cells. The DNA-PEI mixture was incubated for 15 min at room temperature before addition to suspension cells together with VPA at a final concentration of 4 mM. Proteins were expressed for two to three days before the harvest of cells by centrifugation at $500 \times g$. Cell pellets were gently washed with ice-cold PBS, flash-frozen in liquid nitrogen and stored at −80 °C.

Protein purifications were performed at 4 °C unless otherwise stated. For purification of overexpressed homomeric proteins, cell pellets were thawed and proteins were extracted in extraction buffer (25 mM Tris-HCl, pH 8.5, 250 mM NaCl, 50 µg/ml DNase, 10 µM leupeptin, 1 µM pepstatin, 1 µM benzamidine and 2% digitonin) with 75–100 ml buffer per $10 \times g$ of pellet. The mixture was gently rotated for one hour before centrifugation at $10,000 \times g$ for 30 min to remove cell debris. The clarified lysate was mixed with Streptactin Superflow resin (IBA LifeSciences, 0.5 ml resin per 10 ml lysate) and gently stirred for batch binding. After one hour, the slurry was poured into a gravity flow column, and the flow-through was discarded. The resin was washed with 40 column volumes (CVs) of wash buffer (25 mM Tris, pH 8.5, 250 mM NaCl, 0.1% digitonin) before elution of the protein with up to eight CVs of elution buffer (wash buffer containing additional 15 mM D-desthiobiotin). Eluted fractions were pooled and tags were cleaved by incubation with human rhinovirus (HRV) 3 C protease for 60 min. Protein was concentrated to 500 µl with a centrifugal filter with a 100 kDa cut-off and filtered through a 0.22 µm filter. For the initial purification tests of homomeric constructs, the size distribution of purified proteins was assessed by size exclusion chromatography coupled to fluorescence detection (FSEC) (Kawate and Gouaux, 2006) using a Superose 6 5/150 GL column pre-equilibrated in wash buffer. The samples for cryo-EM grid preparation were applied to an AEKTA Prime system for a SEC step using a Superose 6 10/300 GL column (Cytivia) pre-equilibrated in wash buffer. Peak fractions were collected and concentrated using a centrifugal filter with a 100 kDa cut-off and used directly for cryo-EM grid preparation.

For the analysis of the assembly of hLRRC8A with mutants of hLRRC8C, channels were purified from 0.7 l of transfected cells as described above with minor modifications. The extraction buffer contained 25 mM HEPES, pH 7.5, 170 mM NaCl, 3% glycol-diosgenin (GDN), DNase, and protease inhibitors as described above. After the concentration of the eluted fractions, the samples were analyzed by SDS-PAGE with in-gel fluorescence detection or separated on an S6 5/150 GL column connected to a high-pressure liquid chromatography system coupled to a fluorescence detector (Agilent). Absorption and fluorescence spectra were recorded.

For the analysis of the correct channel assembly of hLRRC8C[t379] and hLRRC8C[t419], channels were purified from 35 ml of transfected cells as described above with minor differences. After extraction, the suspension was cleared by centrifugation at $17,000 \times g$ for 40 min. The cleared lysate was filtered through a 0.22-µm filter and subsequently assessed by fluorescence detection (FSEC) based size

exclusion chromatography using a pre-equilibrated Superose 6 5/150 GL column as described above.

## Cryo-EM sample preparation and data collection

Cryo-EM grids of human LRRC8C constructs were frozen from freshly purified proteins at a concentration of 5 mg/ml (hLRRC8C$^{V390L}$), 4.8 mg/ml (LRRC8C$^{trunc}$), and 4.75 mg/ml (LRRC8C$^{WT}$). Holey carbon grids of the type R1.2/1.3 Au 200 mesh (Quantifoil) were glow discharged for 30 s prior to sample application. Freezing was performed using a Vitrobot Mark IV system (ThermoFisher Scientific) at 4 °C and 100% humidity. After application of 2–2.5 μl of sample and removal of excess liquid (by blotting with filter paper for 3–5 s at 0 blotting force), grids were plunge-frozen in a mixture of liquid ethane and propane. Frozen grids were stored in liquid nitrogen until further use.

Cryo-EM datasets were collected on a 300 kV Titan Krios G3i (ThermoFisher Scientific) with a 100 μm objective aperture, a post-column BioQuantum energy filter (Gatan) with a 20 eV slit and a K3 direct electron detector (Gatan). The microscope was operated in super-resolution mode. Dose fractionated micrographs were acquired with a defocus range of −1.0 to −2.4 μm in automated mode using EPU2.9 (ThermoFisher Scientific). Data were acquired at a nominal magnification of 130,000× corresponding to a pixel size of 0.651 Å per pixel (0.3255 Å per pixel in super-resolution) at a total exposure time of 1.26 s (consisting of 47 individual frames) with a dose of ~1.35 e⁻ per Å² per frame. The total electron dose on the specimen level for all datasets was ~65 e⁻/Å² respectively. The structure of hLRRC8C$^{trunc}$, hLRRC8C$^{V390L}$ and hLRRC8C$^{WT}$ were each determined from single datasets containing 20,389, 25,157, and 18,484 micrographs, respectively.

## Cryo-EM image processing

The data processing strategies for LRRC8C$^{trunc}$, LRRC8C$^{V390L}$ and LRRC8C$^{WT}$ are described in detail in Appendix Figs. S1–S3. Datasets were processed using Relion 4.0 (Zivanov et al, 2020) and cryoSPARC v4.3.0. (Punjani et al, 2017). For all datasets, the acquired super-resolution images were gain-corrected and down-sampled twice using Fourier cropping, resulting in a pixel size of 0.651 Å.

For LRRC8C$^{trunc}$ and LRRC8C$^{V390L}$, frames were imported into Relion and used for beam-induced motion correction with a dose-weighting scheme using Relion's own implementation of the MotionCor2 program (Zheng et al, 2017). CTF parameters were estimated using CTFFIND4.1 (Rohou and Grigorieff, 2015). Micrographs showing a large drift, high defocus or poor CTF estimates were removed.

For the LRRC8C$^{trunc}$ dataset, particles were initially manually picked from five micrographs. These particles were used for 2D classification and the best classes were selected as templates for reference-based particle picking from all micrographs. Picked particles were applied to multiple rounds of 2D and 3D classification. Particles from classes showing protein density were imported into cryoSPARC and subjected to an ab initio reconstruction followed by hetero- and non-uniform refinement to generate a reference map. Projections of the obtained map were used for template-based particle picking in cryoSPARC from all original micrographs resulting in 2,517,142 particles that were extracted using a box size of 672 pixels and down-sampled to 168 pixels (pixel size of 2.604 Å/pixel). Particles were cleaned up in several rounds of 2D classification resulting in 639,350 particles that were subjected to an ab initio reconstruction. The two resulting maps were improved by non-uniform and heterogeneous refinement to generate reference densities. A reference map with a protein-free detergent micelle was generated by using 2D class averages without visible protein density in an ab initio reconstruction. The reference maps with and without protein density were used for three steps of heterogeneous refinement, including all initially selected particles to separate empty micelles from micelles containing protein. Particles containing protein density were subjected to a non-uniform refinement step without symmetry applied resulting in a map with a resolution of 7.32 Å. Particles were re-extracted with twofold binning (336-pixel box size, 1.302 Å per pixel), and the non-uniform refinement was repeated with C7 symmetry applied, resulting in a structure at 4.95 Å. This density was further locally refined with a mask around the ESD, resulting in the final reconstruction at a nominal resolution of 3.41 Å.

For the LRRC8C$^{V390L}$ dataset, particles were auto-picked in Relion using templates generated from a previously reported dataset of full-length LRRC8A/C$^{1:1}$/Sb1 (Rutz et al, 2023). Particles were extracted with a box size of 672 pixels and compressed four times (168-pixel box size, 2.604 Å/pixel) for initial processing. The extracted particles were subjected to two rounds of reference-free 2D classification followed by 3D classification with C1 symmetry. As a reference for the 3D classification, a previously determined map of a C7-symmetrized LRRC8C (Rutz et al, 2023) was used after low-pass filtering to 60 Å. Particles showing channels with defined LRR domains were imported into cryoSPARC and subjected to an ab initio reconstruction. Resulting structures showing heptameric LRRC8 channels with different numbers of well-defined subunits were used as references for a heterogenous refinement with all imported particles. Particles of the resulting volumes were further subjected to several rounds of heterogenous and non-uniform refinement to clean-up particles and separate structures with different numbers of visible subunits. Maps showing two, three and four strongly defined subunits and one junk class were used as references for heterogeneous refinement of all initially imported particles, which were initially re-extracted with twofold binning (336-pixel box size, 1.302 Å per pixel). The resulting classes showed channels containing either two or four subunits with strong density including LRRDs (located either next to each other or separated by one or two weakly defined subunits) and one junk class. The three classes with LRRC8 channels were further cleaned up in several heterogeneous refinement steps to separate particles with different numbers of strongly defined subunits. After final rounds of non-uniform refinement, maps at 5.43 Å (containing two well-resolved subunits), 7.57 Å (containing three well-resolved subunits, organized as one pair and an additional isolated subunit) and at 6.62 Å (containing four well-resolved subunits located next to each other) were obtained. The highest resolved map was used for a local refinement step with a mask enclosing protein density at a contour above the threshold of the detergent micelle. Local refinement steps have led to a final map at a resolution of 4.4 Å. This map was used for comparison with the best-resolved map of the LRRC8C$^{WT}$ reconstruction.

For the LRRC8C$^{WT}$ dataset, micrographs were imported into cryoSPARC, subjected to patch motion correction and patch CTF estimation. Low-quality micrographs were discarded based on total full-frame motion distance, relative ice thickness and CTF fit resolution. Particles were picked using template picker with a 30 Å low-pass filtered volume of the C7-symmetrized LRRC8C (Rutz et al, 2023) as a template. Particles were extracted with a box size of 672 pixels and Fourier-cropped to 168 pixels (2.604 Å/pixel). Particles were cleaned up in two rounds of 2D classification. Selected particles were applied to an ab initio reconstruction with two resulting classes that were used as references for a heterogenous refinement step with all particles selected after 2D classification. Particles of the two resulting volumes were further cleaned up in heterogeneous and non-uniform refinement steps. Once the reported resolution reached the Nyquist limit, selected particles were re-extracted with twofold binning (336-pixel box size, 1.302 Å/pixel). Maps reached final resolutions of 7.83 Å and 3.73 Å. The higher resolved map was used for comparison with the map of LRRC8C$^{V390L}$. No symmetry was applied during the reconstruction of LRRC8C$^{V390L}$ and LRRC8C$^{WT}$. The resolution of all generated maps was estimated using a soft solvent mask and based on the gold-standard Fourier Shell Correlation (FSC) 0.143 criterion (Chen et al, 2013; Rosenthal and Henderson, 2003; Scheres and Chen, 2012). The cryo-EM densities were also sharpened using isotropic b-factors.

## Model building and refinement

For structure determination, a previously determined model obtained from a C7-symmetrized LRRC8C$^{WT}$ map (PDB: 8B40) (Rutz et al, 2023) was placed into the respective cryo-EM densities of the best resolve maps of the LRRC8C$^{WT}$, LRRC8C$^{V390L}$ and LRRC8C$^{trunc}$ datasets with Coot (Emsley and Cowtan, 2004). For partially defined subunits, only protein regions with visible density were included in the model. The initial placement was followed by rigid body movement in Phenix treating PDs and LRRDs as separate units followed by few cycles of all atom refinement (Afonine et al, 2018). In case of LRRC8C$^{trunc}$, a model only containing the ESDs was placed into density of the masked map and refined in Phenix maintaining C7 symmetry constraints. The Data collection, refinement and validation statisics is shown in Table EV1. Figures were generated using Chimera (Pettersen et al, 2004), ChimeraX (Pettersen et al, 2021) and Dino (http://www.dino3d.org). Figure 5B was generated with BioRender (BioRender.com).

## Electrophysiology

For electrophysiology, HEK293 LRRC8$^{-/-}$ cells were seeded into Petri dishes at 3% confluency one day before the measurements. Four to five hours after seeding and 14 h before analysis, cells were transfected with a total of 4 µg DNA per 6 cm dish using Lipofectamin 2000 (Invitrogen). The plasmid ratio for hLRRC8A to hLRRC8C constructs (hLRRC8C, hLRRC8C$^{trunc}$ or hLRRC8C$^{V390L}$) and the Venus-only construct was 1:2:2. All measurements were carried out at 20 °C. Patch pipettes were pulled from borosilicate glass capillaries (0.86 mm inner diameter and 1.5 mm outer diameter) with a micropipette puller (Sutter) and fire-polished with a Microforge (Narishige). The typical pipette

resistance was 2–7.5 MΩ. Seals with a resistance of 4 GΩ or higher were used to establish the whole-cell configuration. Data were recorded with an Axopatch 200B amplifier and digitized with a Digidata 1440 or 1550B (Molecular Devices). Analog signals were digitized at 10–20 kHz and filtered at 5 kHz using the built-in four-pole Bessel filter. Data acquisition was performed using the Clampex 10.7 software (Molecular Devices). Two buffer systems with distinct hypotonic buffers for the induction of cell swelling were used for the measurements. Both induced robust VRAC currents in HEK293 cells in response to the change to hypotonic conditions (Sukalskaia et al, 2021; Voss et al, 2014). In buffer system 1 reducing the salt concentration in hypotonic conditions (Δsalt), which was previously used by Voss et al (Voss et al, 2014), the pipette buffer contained 10 mM HEPES, pH 7.4, 40 mM CsCl, 100 mM Cs-Met-sulfonate, 1 mM MgCl$_2$, 1.9 mM CaCl$_2$, 5 mM EGTA, 4 mM Na$_2$ATP (osmolarity of 290 mmol/kg), the isotonic bath solution contained 10 mM HEPES, pH 7.4, 150 mM NaCl, 6 mM KCl, 1 mM MgCl$_2$, 1.5 mM CaCl$_2$, 10 mM glucose (osmolarity of 320 mmol/kg) and the hypotonic bath solution 10 mM HEPES, pH 7.4, 105 mM NaCl, 6 mM KCl, 1 mM MgCl$_2$, 1.5 mM CaCl$_2$, 10 mM glucose (osmolarity of 240 mmol/kg). In buffer system 2, reducing the mannitol concentration in hypotonic conditions (Δmannitol), (Han et al, 2019; Sukalskaia et al, 2021)), the pipette buffer contains 10 mM HEPES, pH 7.4, 150 mM NMDG-Cl, 1 mM EGTA, 2 mM Na$_2$ATP (osmolarity of 280 mmol/kg), the isotonic bath solution 10 mM HEPES, pH 7.4, 95 mM NaCl, 1.8 mM CaCl$_2$, 0.7 mM MgCl$_2$, 100 mM mannitol (osmolarity of 300 mmol/kg) and the hypotonic bath solution contains 10 mM HEPES, pH 7.4, 95 mM NaCl, 1.8 mM CaCl$_2$, 0.7 mM MgCl$_2$ (osmolarity of 200 mmol/kg) (Sukalskaia et al, 2021). Currents from transfected hLRRC8A/C, hLRRC8A/C$^{trunc}$ or hLRRC8A/C$^{V390L}$ channels were measured in both buffer systems. Cells were perfused locally using a gravity-fed system. Only cells with Venus fluorescence were selected for recording. After break-in into the cell and establishing the whole-cell configuration, cells were perfused with isotonic bath buffer. After one minute, cell swelling was initiated by changing the perfusion from isotonic to hypotonic buffer. Currents were monitored at 2 s intervals for 6–7 min using a ramp protocol (15 ms at 0 mV, 100 ms at −100 mV, a 500-ms linear ramp from −100 mV to 100 mV, 100 ms at 100 mV, 200 ms at −80 mV, 1085 ms at 0 mV). Values at 100 mV, 10 ms after the ramp, are shown in the activation curves. Current–voltage (I–V) relationships were obtained from a voltage step protocol (from −100 to 120 mV in 20 mV steps). After the voltage step protocol, cells were perfused with hypotonic buffer for an additional 20–30 s before switching to isotonic solution to initiate cell shrinkage. Current inactivation was monitored using the same ramp protocol as described above. Only one cell per dish was used for measurements in hypotonic conditions. For the analysis of currents from the transfected hLRRC8A/C, hLRRC8A/C$^{trunc}$ and hLRRC8A/C$^{V390L}$ channels, the recordings from all measured cells that did not show leakage were included in the analysis and are displayed in the corresponding figures. Leakage was detected for all constructs by a weak and instable seal and by the characteristics of current–voltage relationships including poor selectivity based on the chloride reversal potential, weak outward rectification and absent inactivation at strongly positive voltage. On this basis, the number of discarded recordings were comparable for all investigated constructs in both conditions. Basal currents of

LRRC8C mutants recorded at isotonic conditions displayed the described characteristic VRAC features (Fig. EV2B,D). Selectivity properties of endogenous and activated currents were recorded as described in buffer system Δsalt (Lutter et al, 2017). To measure shifts in reversal potentials, NaCl in isotonic and hypotonic buffers was replaced with equimolar amounts of NaI and Na-glutamate. For investigation of selectivity properties under isotonic conditions the cells were initially perfused with isotonic buffer containing NaCl for 60 s after establishment of the whole-cell configuration before changing to isotonic buffer containing NaI. After 60 s perfusion, the buffer was changed to isotonic buffer containing Na-glutamate and perfused for another 60 s. During perfusion, currents were monitored by ramp protocols while reversal potentials were obtained from current–voltage relationships recorded by a step protocol as described above. To evaluate selectivity properties after activation, the initial isotonic perfusion and activation by hypotonic perfusion was carried out in buffer system Δsalt. After the perfusion with hypotonic buffer containing NaCl, the buffer was changed to hypotonic buffer containing NaI. After 60 s perfusion, the buffer was changed to hypotonic buffer containing Na-glutamate. As described for selectivity experiments under isotonic conditions, currents during activation were monitored through ramp protocols and reversal potentials were calculated from current–voltage relationship measurements recorded by step protocols. All recordings used to derive reversal potentials displayed in Fig. EV2F did not show leakage and displayed characteristic VRAC features such as outward rectification and channel inactivation at positive voltages.

Inhibition of endogenous currents of hLRRC8A/C$^{\text{trunc}}$ and hLRRC8A/C$^{\text{V390L}}$ channels was investigated by perfusion of the VRAC inhibitor DCPIB (Best et al, 2010; Yamada et al, 2021) in buffer Δsalt. After the establishment of whole-cell configuration in transfected cells expressing hLRRC8A/C$^{\text{trunc}}$ and hLRRC8A/C$^{\text{V390L}}$ channels, cells were perfused with isotonic buffer for 60 s before changing to isotonic buffer supplemented with 50 μM DCPIB. The current decrease was recorded continuously by a ramp protocol and the effect of the inhibitor was quantified as the ratio of currents recorded after 30 s of perfusion with DCPIB buffer to currents recorded after 30 s of perfusion with unsupplemented isotonic buffer prior to the addition of DCPIB.

The relation between current densities measured from cells expressing WT or mutated LRRC8C subunits was analyzed by Brown–Forsythe and Welch's ANOVA tests with Dunnett's T3 multiple comparisons testing. The relation between currents of the same constructs recorded from isotonic and hypotonic solutions was analyzed by unpaired *t* tests with applied Welsch's correction. Data were analyzed using Clampfit 10.6 (Molecular Devices), GraphPad Prism 10.0.2 and Excel (Microsoft).

## Data availability

All relevant genetic data produced in this work are presented within the text, figures or tables. Additional genotypes are available upon reasonable request, and could be released only if in agreement with current regulations on Individual Data Protection. The three-dimensional cryo-EM density maps have been deposited in the Electron Microscopy Data Bank under accession numbers EMD-19495 (hLRRC8C), EMD-50123 (hLRRC8C$^{\text{V390L}}$), EMD-50072

(hLRRC8C$^{\text{trunc}}$, ESD, C7 symmetrized), EMD-50073 (hLRRC8C$^{\text{trunc}}$, full-length, C1 symmetrized), EMD-50074 (hLRRC8C$^{\text{trunc}}$, full-length, C7 symmetrized). The deposition includes maps and the corresponding two half-maps. Coordinates have been deposited in the Protein Data Bank under accession numbers 8RTS (hLRRC8C), 9F16 (hLRRC8C$^{\text{V390L}}$) and 9EZC (hLRRC8C$^{\text{trunc}}$, ESD, C7 symmetrized). Electrophysiological data that support the findings of this study are provided as source data file.

The source data of this paper are collected in the following database record: biostudies:S-SCDT-10_1038-S44318-024-00322-y.

## Peer review information

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

## Acknowledgements

This research was supported by grants from the Swiss National Science Foundation to RD (No. 310030_204373). The authors thank the Center for Microscopy and Image Analysis (ZMB) of the University of Zurich for access to the electron microscopes and T.J. Jentsch for providing the *LRRC8*$^{-/-}$ HEK cell line. All members of the Dutzler lab and the Rivolta lab are acknowledged for their help at various stages of the project.

## Author contributions

**Mathieu Quinodoz**: Data curation; Formal analysis; Validation; Investigation; Visualization; Writing—original draft; Writing—review and editing; Conceived and coordinated the molecular DNA studies and the bioinformatic evaluations.
**Sonja Rutz**: Data curation; Formal analysis; Validation; Investigation; Visualization; Writing—original draft; Writing—review and editing; Expressed and purified proteins, prepared samples for cryo-EM, collected cryo-EM data, proceeded with structure determination and refinement and recorded electrophysiology data and analyzed and interpreted these data. **Virginie Peter**: Data curation; Formal analysis; Investigation; Writing—review and editing; Conceived and coordinated the molecular DNA studies and the bioinformatic evaluations. **Livia Garavelli**: Data curation; Investigation; Writing—original draft; Writing—review and editing; Evaluated the patients and coordinated the diagnostic procedures and provided the description of clinical data.
**A Micheil Innes**: Data curation; Investigation; Writing—original draft; Writing—review and editing; Evaluated the patients and coordinated the diagnostic procedures and provided the description of clinical data. **Elena F Lehmann**: Data curation; Formal analysis; Validation; Investigation; Visualization; Writing—review and editing; Expressed and purified proteins, recorded electrophysiology data and analyzed and interpreted these data.
**Stephan Kellenberger**: Writing—review and editing; Performed initial electrophysiology studies in cell cultures. **Zhong Peng**: Writing—review and editing; Performed initial electrophysiology studies in cell cultures.
**Angelica Barone**: Writing—review and editing; Evaluated the patients and coordinated the diagnostic procedures. **Belinda Campos-Xavier**: Writing—review and editing; Reproduced and validated the DNA sequencing data.
**Sheila Unger**: Writing—review and editing; Evaluated the patients and coordinated the diagnostic procedures. **Carlo Rivolta**: Conceptualization; Data curation; Formal analysis; Supervision; Funding acquisition; Writing—original draft; Project administration; Writing—review and editing; Conceived and coordinated the molecular DNA studies and the bioinformatic evaluations.
**Raimund Dutzler**: Conceptualization; Supervision; Funding acquisition; Writing—original draft; Project administration; Writing—review and editing.
**Andrea Superti-Furga**: Conceptualization; Supervision; Funding acquisition; Writing—original draft; Project administration; Writing—review and editing; Evaluated the patients and coordinated the diagnostic procedures. Conceived and coordinated the molecular DNA studies and the bioinformatic evaluations.

Source data underlying figure panels in this paper may have individual authorship assigned. Where available, figure panel/source data authorship is listed in the following database record: biostudies:S-SCDT-10_1038-S44318-024-00322-y.

## Disclosure and competing interests statement

The parents of the two patients in this study have been fully informed that the images shown would be used in a scientific research publication, and that the children will be fully identifiable in this publication, and given their consent. Showing such identifiable patient photographs is necessary for future clinical diagnosis in other affected children of the monoallelic variants of the LRRC8C gene similar to those reported here. We thankfully acknowledge the graciousness and generosity of the parents in giving this consent in the interest of helping future diagnoses in other affected children and in fostering knowledge about and research in this condition. The authors declare no competing interests.

# Expanded View Figures

**Figure EV1. Features of the hLRRC8C mutant V390L.**

Comparison of the highest resolved cryo-EM maps (contoured at 8σ) obtained from the hLRRC8C$^{V390L}$ and hLRRC8C$^{WT}$ datasets. (A) Cryo-EM map of hLRRC8C$^{V390L}$ (top) and hLRRC8C$^{WT}$ (bottom) superimposed on indicated subunits. (B) hLRRC8C$^{V390L}$ (left) and hLRRC8C$^{WT}$ (right) densities viewed from the cytoplasm. (A, B) Parts of protein subunits that are defined in the density are shown as ribbons, subunit numbering is as in Fig. 3D, E.

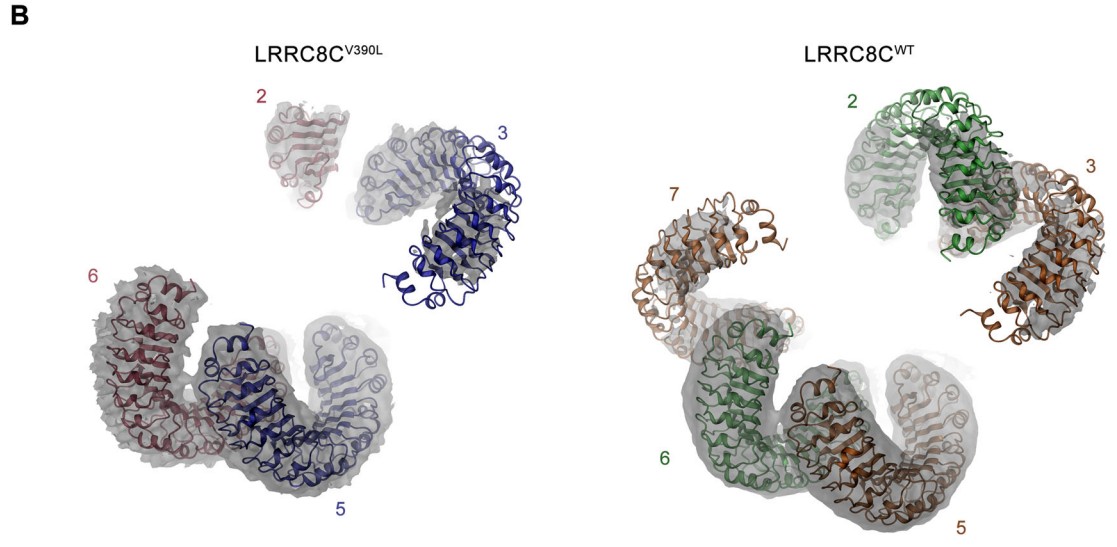

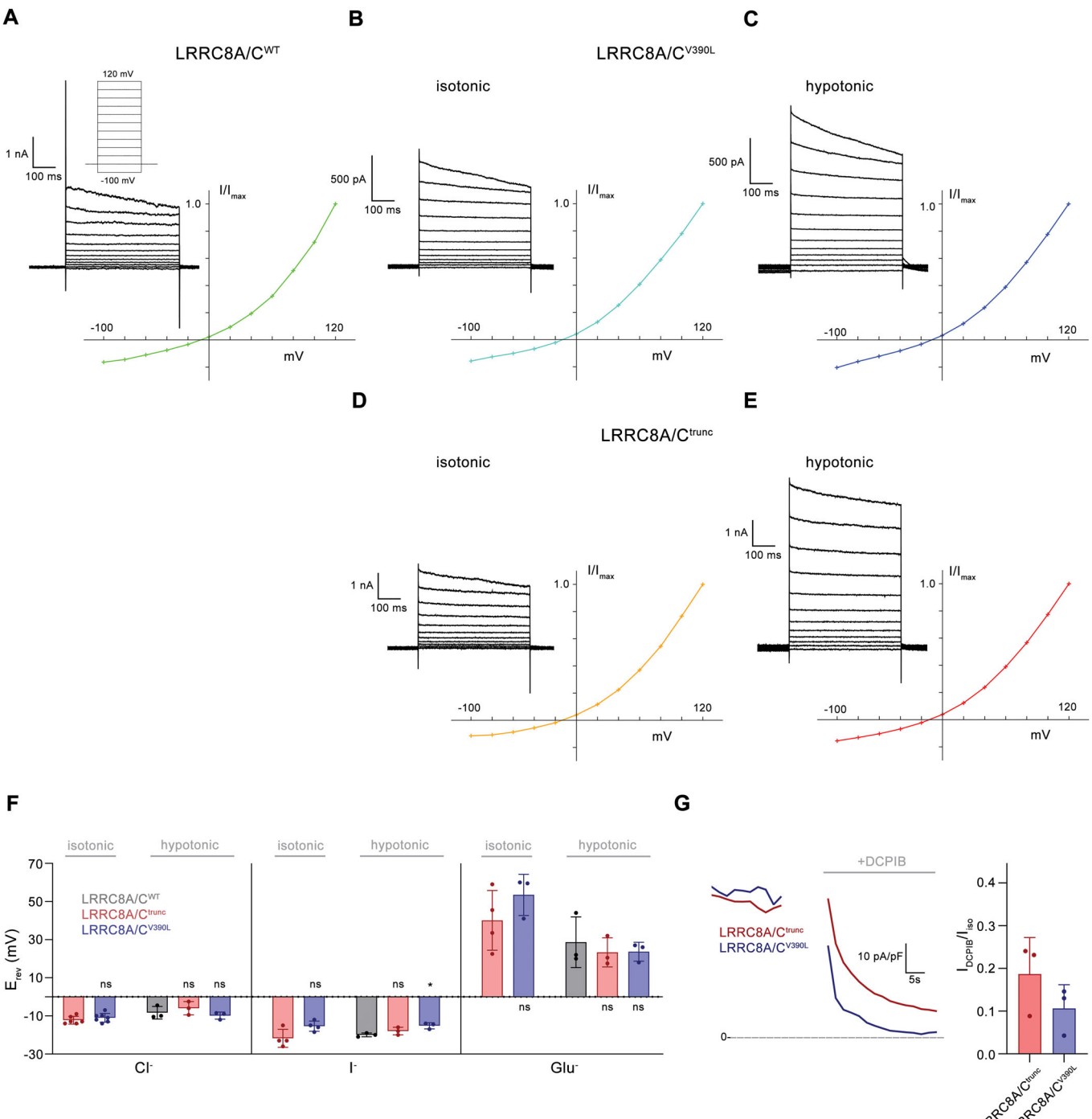

◀

**Figure EV2. Functional properties of hLRRC8C mutants in buffer set Δsalt.**

(A) Representative current traces and current–voltage relationship of an LRRC8$^{-/-}$ cell transfected with constructs coding for hLRRC8A and hLRRC8C$^{WT}$ after swelling of cells in buffer set Δsalt showing the expected features of an LRRC8A/C channel. (B, C) Representative current traces and current–voltage relationships of an LRRC8$^{-/-}$ cell transfected with constructs coding for hLRRC8A and hLRRC8C$^{V390L}$ under isotonic (B) and hypertonic conditions (C). (D, E) Representative current traces and current–voltage relationships of an LRRC8$^{-/-}$ cell transfected with constructs coding for hLRRC8A and hLRRC8C$^{trunc}$ under isotonic (D) and hypertonic conditions (E). The similar general properties of currents obtained under isotonic and hypotonic conditions with respect to the negative reversal potential, the pronounced outward rectification and the inactivation at strongly positive voltages show characteristics of VRACs composed of LRRC8A and C-subunits. (A–E) Currents were recorded by patch-clamp electrophysiology in the whole-cell configuration upon perfusion of cells with either isotonic (B, D) or hypotonic buffer (A, C, E, described in "Methods"). Normalized IV-plots are displayed (right, bottom). Voltage protocol is as shown in (A). (F) Reversal potentials of currents mediated by LRRC8A/C$^{trunc}$ or LRRC8A/C$^{V390L}$ in isotonic and hypotonic conditions of buffer system 1 (Δsalt) in comparison to LRRC8A/C$^{WT}$ currents measured in hypotonic conditions of the same buffer system. Cl$^{-}$ refers to standard solutions, I$^{-}$ and Glu$^{-}$ to solutions where NaCl was replaced by equimolar amounts of NaI or NaGlu, respectively. Due to the lower anion concentration, reversal potentials are generally lower in hypotonic solutions. Panels show measurements of individual cells (spheres), bars represent mean currents, errors are s.e.m. Asterisks indicate significant deviations from WT in the respective conditions in a Brown–Forsythe and Welch's ANOVA test with Dunnett's T3 multiple comparison testing in case of the comparison of currents in hypotonic conditions and deviations from LRRC8C$^{trunc}$ in a unpaired $t$ test with Welch's correction in case of isotonic conditions. Differences were usually not statistically significant (ns) except for one case (*$P = 0.0375$). (G) Inhibition of endogenous currents of LRRC8A/C$^{trunc}$ and LRRC8A/C$^{V390L}$ in response to perfusion of isotonic solutions of buffer condition Δsalt containing 50 μM DCPIB. Left, current decrease in representative traces in response to inhibitor addition measured at 100 mV by a ramp protocol. Dashed line refers to zero current. Right, fraction of inhibition of basal currents from respective constructs in response to DCPIB application. Measurements of individual cells are shown as spheres, bars represent mean currents, errors are s.e.m.

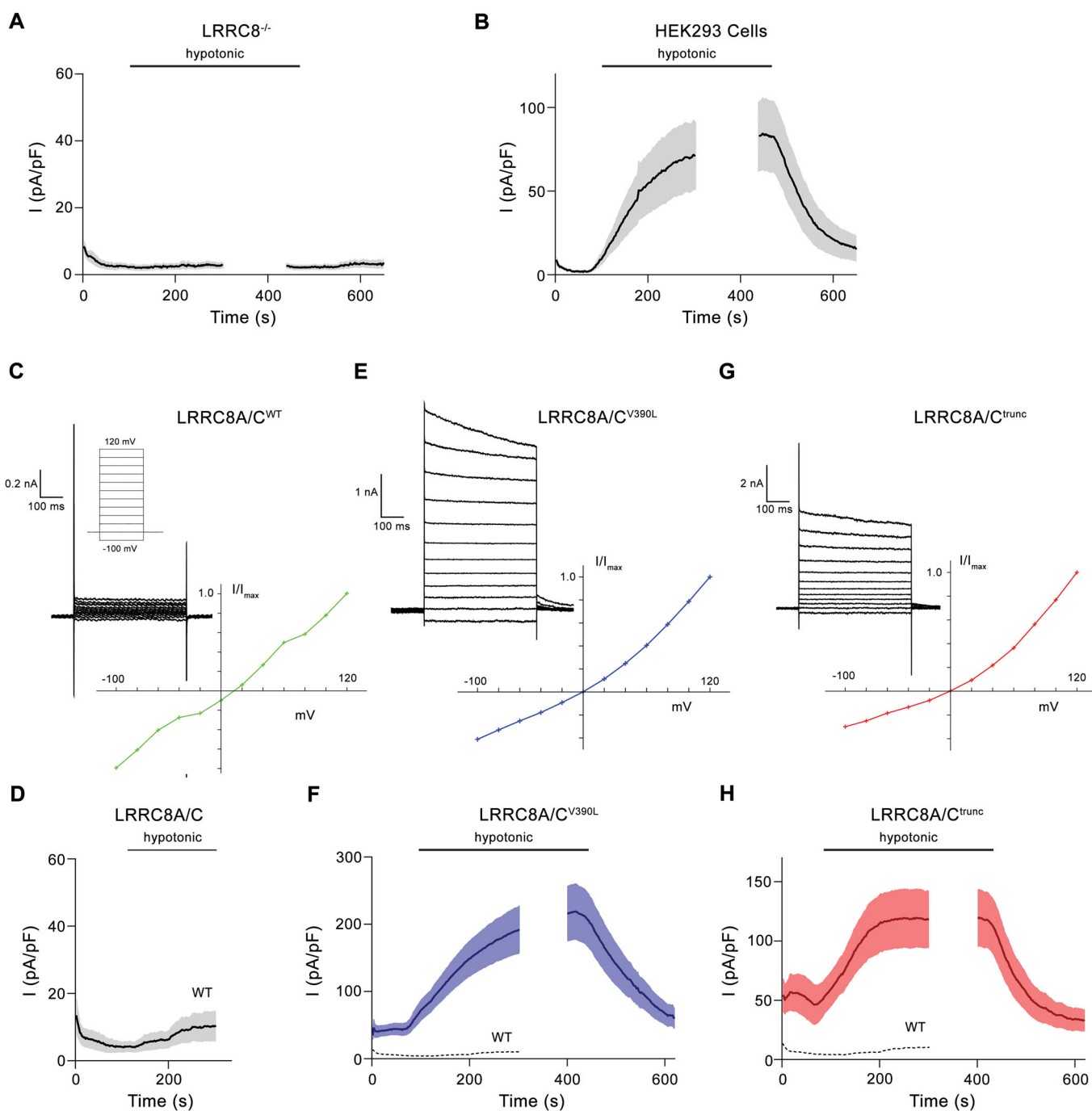

**Figure EV3. Functional properties of hLRRC8C mutants in buffer set Δmannitol.**

(A, B) Mean current density (at 100 mV) of LRRC8A$^{-/-}$ cells ($n = 10$) (A) and HEK293 cells ($n = 8$) (B) recorded in buffer set Δmannitol. (C) Representative current traces and current–voltage relationship of LRRC8$^{-/-}$ cells transfected with constructs coding for hLRRC8A and hLRRC8C$^{WT}$ after swelling of cells in buffer set Δmannitol not leading to channel activation. (D) Mean current density (at 100 mV) of hLRRC8A/C$^{WT}$ channels ($n = 12$) recorded in buffer set Δmannitol. (E) Representative current traces and current–voltage relationship of LRRC8$^{-/-}$ cells transfected with constructs coding for hLRRC8A and hLRRC8C$^{V390L}$ after swelling of cells in buffer set Δmannitol. (F) Mean current density (at 100 mV) of hLRRC8A/C$^{V390L}$ ($n = 11$) recorded in buffer set Δmannitol. (G) Representative current traces and current–voltage relationship of LRRC8$^{-/-}$ cells transfected with constructs coding for hLRRC8A and hLRRC8C$^{trunc}$ after swelling of cells in buffer set Δmannitol. (H) Mean current density (at 100 mV) of hLRRC8A/C$^{trunc}$ channels ($n = 10$) recorded in buffer set Δmannitol. (A–H) Currents were recorded by patch-clamp electrophysiology in the whole-cell configuration. Cannels were activated by perfusion of cells with hypotonic buffers (described in methods). (C, E, G) Normalized IV-plots are displayed (right, bottom). Voltage protocol is as shown in (C). (F, H) The mean current of WT is shown as dashed line for comparison. (D, F, H) Panels show mean of indicated number of biological replicates, errors are s.e.m. Note the different scale of the y-axis in the three panels.

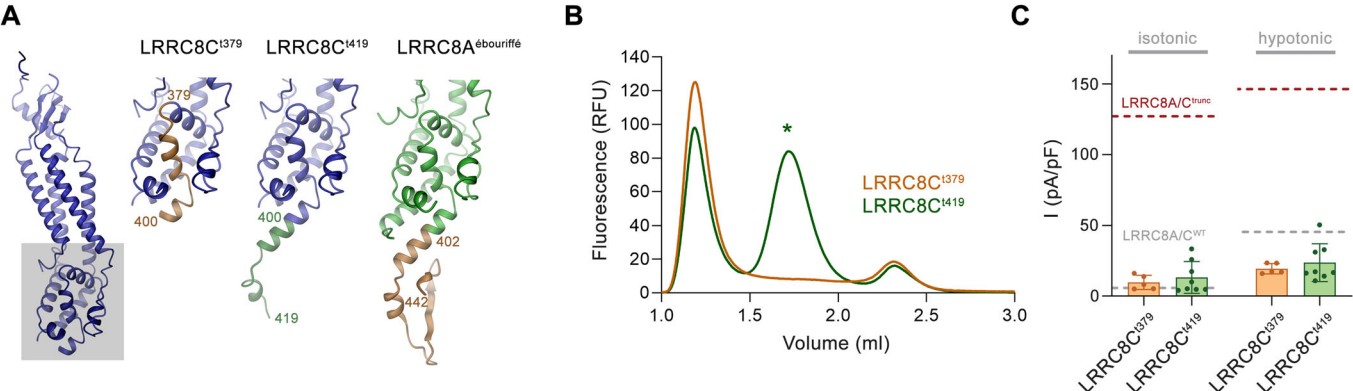

**Figure EV4.  Biochemical and functional characterization of the constructs LRRC8C^t379 and LRRC8C^t419.**

(A) Models of truncated LRRC8 constructs. A ribbon representation of LRRC8C^trunc is shown left as reference. The gray box indicates the region of interest in different constructs shown to the right: Center left, LRRC8C^t379, the region on the C-terminus of LRRC8C^trunc (380–400) that is absent in this construct is highlighted in orange; Center right, LRRC8C^t419, the sequence extension compared to LRRC8C^trunc (residues 401–419) is highlighted in green; Right, equivalent region of the LRRC8A mutant found in éburiffé mice (which is truncated after residue 442), with the extension compared to LRRC8C^trunc highlighted in orange. (B) Fluorescence size exclusion chromatogram of GFP-fusions of the constructs LRRC8C^t379 and LRRC8C^t419 after extraction in detergent. Asterisk corresponds to the volume of the heptameric channel. While no oligomer is detected in case of the shorter LRRC8C^t379, the longer LRRC8C^t419 elutes at a similar volume as LRRC8C^trunc. (C) Mean current density (at 100 mV) of LRRC8^-/- cells transfected with constructs coding for LRRC8A and either LRRC8C^t419 ($n = 5$) or LRRC8C^t419 ($n = 8$) recorded by patch-clamp electrophysiology in the whole-cell configuration in isotonic (measured 10 s after establishment of whole-cell configuration) and hypotonic conditions (measured 4 min after switch to hypotonic solutions) in buffer system 1 (Δsalt). Panels show measurements of individual cells (spheres), bars represent mean currents, errors are s.e.m. Mean current densities obtained from LRRC8A/C^WT (red) and LRRC8A/C^trunc channels (blue, obtained from Fig. 4G) are shown as dashed line for comparison. Deviations of currents of both constructs to WT and mutated LRRC8C in the respective conditions were evaluated by a Brown–Forsythe and Welch's ANOVA test with Dunnett's T3 multiple comparison testing. Deviations between isotonic and hypotonic conditions for each construct was separately evaluated by unpaired $t$ testing with Welsch's correction applied. Differences to LRRC8A/C^WT were in all cases found to be not statistically significant (LRRC8A/C^t379 isotonic, $P = 0.542$, hypotonic, $P = 0.302$; LRRC8A/C^t419 isotonic, $P = 0.236$, hypotonic, $P = 0.366$). Differences to LRRCA/C^trunc were evaluated as significant (LRRC8A/C^t379 isotonic, $P = 0.0023$, hypotonic, $P = 0.0028$; LRRC8A/C^t419 isotonic, $P = 0.0002$, hypotonic, $P = 0.0003$). Current increases of both constructs upon change to hypotonic conditions were evaluated as statistically not significant (LRRC8A/C^t379 $P = 0.274$; LRRC8A/C^t419, $P = 0.113$).

