## [Peer Review File · The EMBO Journal]

De novo variants in LRRC8C resulting in constitutive channel activation cause a human multisystem disorder

Mathieu Quinodoz, Sonja Rutz, Virginie Peter, Livia Garavelli, Micheil Innes, Elena Lehmann, Stephan Kellenberger, Zhong Peng, Angelica Barone, Belinda Campos-Xavier, Sheila Unger, Carlo Rivolta, Raimund Dutzler, and Andrea Superti-Furga

Corresponding author(s): Raimund Dutzler (dutzler@bioc.uzh.ch) , Andrea Superti-Furga (asuperti@unil.ch), Carlo Rivolta (carlo.rivolta@iob.ch)

Review Timeline:

Submission Date:	2nd Apr 24
Editorial Decision:	15th May 24
Revision Received:	23rd Aug 24
Accepted:	8th Nov 24

Editor: William Teale

Transaction Report:

Dear Raimund,

Thank you again for the submission of your manuscript entitled "Mutations in LRRC8C resulting in constitutive channel activation cause a multisystem disorder" (117476) and for your patience during the review process. We have now received three reports from the referees, which I copy below.

As you can see from their comments, each referees raise a number of concerns that will require your attention before your manuscript can be published in The EMBO Journal.

Based on the overall interest expressed in the reports, though, I would like to invite you to address the comments of all referees in a revised version of the manuscript. I should add that it is The EMBO Journal policy to allow only a single major round of revision and that it is therefore important to resolve the main concerns at this stage. I believe the concerns of the referees are reasonable and addressable, but I remain available for a Zoom discussion if you would like to discuss the referee comments or if you anticipate any problems in addressing any of their points. Please, follow the instructions below when preparing your manuscript for resubmission.

I would also like to point out that as a matter of policy, competing manuscripts published during this period will not be taken into consideration in our assessment of the novelty presented by your study ("scooping" protection). We have extended this 'scooping protection policy' beyond the usual 3 month revision timeline to cover the period required for a full revision to address the essential experimental issues. Please contact me if you see a paper with related content published elsewhere to discuss the appropriate course of action.

Again, please contact me at any time during revision if you need any help or have further questions.

Thank you very much again for the opportunity to consider your work for publication. I look forward to your revision.

Best regards,

Will

William Teale, Ph.D.
Editor
The EMBO Journal

When submitting your revised manuscript, please carefully review the instructions below and include the following items:

- 1) a .docx formatted version of the manuscript text (including legends for main figures, EV figures and tables). Please make sure that the changes are highlighted to be clearly visible.
- 2) individual production quality figure files as .eps, .tif, .jpg (one file per figure).
- 3) a .docx formatted letter INCLUDING the reviewers' reports and your detailed point-by-point response to their comments. As part of the EMBO Press transparent editorial process, the point-by-point response is part of the Review Process File (RPF), which will be published alongside your paper.
- 4) a complete author checklist, which you can download from our author guidelines ([https://wol-prod-cdn.literatumonline.com/pb-assets/embo-site/Author Checklist%20-%20EMBO%20J-1561436015657.xlsx](https://wol-prod-cdn.literatumonline.com/pb-assets/embo-site/Author%20Checklist%20-%20EMBO%20J-1561436015657.xlsx)). Please insert information in the checklist that is also reflected in the manuscript. The completed author checklist will also be part of the RPF.
- 5) Please note that all corresponding authors are required to supply an ORCID ID for their name upon submission of a revised manuscript.
- 6) We require a 'Data Availability' section after the Materials and Methods. Before submitting your revision, primary datasets produced in this study need to be deposited in an appropriate public database, and the accession numbers and database listed under 'Data Availability'. Please remember to provide a reviewer password if the datasets are not yet public (see <https://www.embopress.org/page/journal/14602075/authorguide#datadeposition>). If no data deposition in external databases is

needed for this paper, please then state in this section: This study includes no data deposited in external repositories. Note that the Data Availability Section is restricted to new primary data that are part of this study.

Note - All links should resolve to a page where the data can be accessed.

8) For data quantification: please specify the name of the statistical test used to generate error bars and P values, the number (n) of independent experiments (specify technical or biological replicates) underlying each data point and the test used to calculate p-values in each figure legend. The figure legends should contain a basic description of n, P and the test applied. Graphs must include a description of the bars and the error bars (s.d., s.e.m.).

9) We would also encourage you to include the source data for figure panels that show essential data. Numerical data can be provided as individual .xls or .csv files (including a tab describing the data). For 'blots' or microscopy, uncropped images should be submitted (using a zip archive or a single pdf per main figure if multiple images need to be supplied for one panel). Additional information on source data and instruction on how to label the files are available at .

10) We replaced Supplementary Information with Expanded View (EV) Figures and Tables that are collapsible/expandable online (see examples in <https://www.embopress.org/doi/10.15252/embj.201695874>). A maximum of 5 EV Figures can be typeset. EV Figures should be cited as 'Figure EV1, Figure EV2" etc. in the text and their respective legends should be included in the main text after the legends of regular figures.

12) Our journal encourages inclusion of *data citations in the reference list* to directly cite datasets that were re-used and obtained from public databases. Data citations in the article text are distinct from normal bibliographical citations and should directly link to the database records from which the data can be accessed. In the main text, data citations are formatted as follows: "Data ref: Smith et al, 2001" or "Data ref: NCBI Sequence Read Archive PRJNA342805, 2017". In the Reference list, data citations must be labeled with "[DATASET]". A data reference must provide the database name, accession number/identifiers and a resolvable link to the landing page from which the data can be accessed at the end of the reference. Further instructions are available at .

Further instructions for preparing your revised manuscript:

- a point-by-point response to the referees' comments, with a detailed description of the changes made (as a word file).
 - a word file of the manuscript text.
 - individual production quality figure files (one file per figure)
 - a complete author checklist, which you can download from our author guidelines (<https://www.embopress.org/page/journal/14602075/authorguide>).
 - Expanded View files (replacing Supplementary Information)
- Please see out instructions to authors
<https://www.embopress.org/page/journal/14602075/authorguide#expandedview>

We realize that it is difficult to revise to a specific deadline. In the interest of protecting the conceptual advance provided by the work, we recommend a revision within 3 months (13th Aug 2024). Please discuss the revision progress ahead of this time with the editor if you require more time to complete the revisions. Use the link below to submit your revision:

Referee #1:

In this very interesting paper Quinodoz et al. convincingly establish a causal link between de novo variants in LRRC8C and a clinical phenotype characterized by telangiectasia, microcephaly, metaphyseal dysplasia, eye abnormalities and short stature. This is a major step forward, as the authors are the first to identify mutations in LRRC8s in human pathology in the decade after the molecular identification of LRRC8 proteins as constituents of VRAC. The authors employ cryo-electron microscopy, biochemical techniques and patch-clamp electrophysiology, to show that the identified mutations introduce more flexibility to the heteromeric channel and hence mediate higher ion currents. The data appear of high quality and the writing is generally clear.

I have no major points.

Minor:

1. In order to influence channel function, LRRC8Ctrunc and LRRC8CV390L have to assemble with VRACs essential LRRC8A subunit. The authors perform a pull-down experiment (Fig.4 A, B and C) in which they show binding of the two LRRC8C variants to LRRC8A. The trustworthiness of these results could be strengthened by including a control condition in which the cells express only LRRC8A-GFP in the absence of LRRC8C-SBP. In this way unspecific binding of LRRC8A-GFP to the column could be ruled out. We think that this is a minor point since the authors confirm that LRRC8Ctrunc LRRC8CV390L form functional channels with LRRC8A (Fig.4 E-H) for which assembly of the subunits is a prerequisite.
2. The "Impact of disease mutations on LRRC8A/C channel function" section does not read as self-contained: two different buffer combinations are mentioned, but the motivation to use both in parallel nor difference between them was explained. The manuscript would read more smoothly if it were stated that e.g. "in C1, the concentration of NaCl is reduced to lower the extracellular osmolarity, while to this end in C2, mannitol is omitted, while ion concentrations remain constant". It is also puzzling that in Sukalskaia et al, 2021 C2 caused activation of overexpressed LRRC8A/C channels in LRRC8^{-/-} background, albeit the currents were smaller than those in WT HEK cells, but in the present manuscript, ICl_{vol} was seen only upon overexpression of mutant channels in these ionic conditions. What can be the reason for such differences?
3. Although the identified mutations are located far from the pore region, there are indications that strong VRAC activation may dampen its selectivity. Especially the truncating mutation is reminiscent of the result of proteolytic cleavage of the distantly related pannexin1 channels that allows for ATP transport (<https://doi.org/10.1038/s41586-020-2357-y>). It would be interesting to check whether the identified mutations affect halide permeability ratios or even facilitate permeation of organic substrates such as glutamate or taurine.
4. It would be nice to compare the effects of LRRC8Ctrunc to a spontaneous mouse LRRC8A mutation ébouriffé that truncates the last 15 LRRs.
5. The exact region where the mutations are located should be stated in the abstract.
6. In line 161 of the manuscript is a typing error. The authors refer to Figure EV3 which is actually EV4

Referee #2:

The manuscript describes two unrelated children with a de novo mutation in the LRR8C channel, one with a missense (p.(Val390Leu)) and one frameshift mutation (p.(Leu400IlefsTer8)). These specific variants are not present in GnomAD and fall in a highly conserved region of the protein, which also is intolerant to variation in the normal population. The children show a remarkable similarity in phenotypes of blood vessels, brain, eye and bones. In addition, the two children share a similar facial gestalt. LRR8C has not before linked to a human syndrome. The authors propose to call this disorder LRR8C-related TIMES Syndrome for telangiectasia, microcephaly, metaphyseal dysplasia, eye abnormalities and short stature.

Expression studies of wt and mutated channels in HEK293 suggested proper folding and assembly of both mutants into the same monomeric heptameric state as WT. Cryo-EM studies of wt and mutants showed an increased flexibility of the membrane-inserted pore domain for both the mutants compared to the wildtype. Cellular function of LRR8 channels is believed to work via heterodimers between LRR8A and LRR8C subunits. Next the authors convincingly show that both mutants could still be copurified as a heterodimer. Patch clamp experiments in LRR8 knockout HEK293 cells transfected with mutants and wt LRR8C showed an overactivity of the two mutant channels.

This manuscript describes the first human syndrome associated with mutations in LRR8C, thus providing an important contribution to the field of human disease genetics. The description of LRR8C mutations give insight in its function in the cell and during development. The functional studies show a mechanistic explanation of the gain-of-function of the mutated channels. These findings are of importance for both clinical genetics as well as provide important novel biological insights.

Major:

1. Two patients with de novo mutations in the same gene to define a novel syndrome is the absolute minimum, having (an) additional patient(s) would strengthen the manuscript. However, I see that the rarity of gain-of-function mutations might complicate this. The different types of mutation leading to the similar and specific phenotypes in both patients is a plus here.
2. The patient mutations fall in a highly conserved region. The authors state that within this region the rate of missense changes is ~8-fold lower than the rest of the gene, and by the complete absence of frameshift or nonsense variants. I agree with this observation. The width of this conserved region, however, is narrow, as GnomAD lists truncating alleles in the normal population (e.g. around amino acid position 400: p.Trp340Ter, p.Glu357ArgfsTer10, p.Phe370CysfsTer11, p.Gln379Ter, p.Gln419Ter, p.Arg425Ter). In the current manuscript it is not clear how truncation at position 400 leads to so such diametrically different phenotypes compared to truncation at 379 and 419. These or similar variants in the population should be included in the functional experiments, as negative controls, to provide further evidence of the specific gain-of-function effects of the patient mutations.
3. Similar reasoning as comment 2 holds true for the missense variants present in the population, although additional functional experiments for the truncating variants would suffice in a revision.
4. The discussion reads 'Haploinsufficiency, i.e. pathogenicity due to mutation-induced insufficient protein production, can be excluded as 220 control individuals carrying loss-of-function variants in exon 2 have been reported in the gnomAD database (v4.0) (Karczewski et al., 2020).' This contradicts the fact that GnomAD shows a significantly lower observed over expected ratio for both missense and nonsense variants in LRR8C. Please discuss this in the manuscript.

Minor:

1. Please refrain from using the term mutation (abstract). The term "sequence variation" is used to prevent confusion with the term "mutation" and "polymorphism", mutation meaning "change" in some disciplines and "disease-causing change" in others and polymorphism meaning "non disease-causing change" or "change found at a frequency of 1% or higher in the population".
2. Please provide genomic mutation information of the two mutations (hg38)
3. Clinical geneticists would be helped with a (supplemental) table summarizing the clinical features for each patient
4. Line 989: typo: Iso
5. Line 998: typo: bléomycin

Referee #3:

In this manuscript, Mathieu Quinodoz and colleagues identified two de novo mutations in LRR8C gene in two similar cases of a congenital syndrome affecting multiple systems. Based on structural and electrophysiological analysis on the two mutants, the authors propose that the mutations impair VRAC channel gating and result in gain-of-function, which likely underlying the disease pathogenesis. The findings are novel and significant because it represents the first clear evidence linking VRAC/LRR8C

channel mutations to human pathology. It also suggests the linker region between the pore domain and LRR domains as a critical element in VRAC channel gating. Here are a few major comments:

1. Because the valine to leucine mutation is very subtle, it would add a lot of confidence to the authors' conclusion if similar mutations in LRRC8A and other LRRC8 genes, particular LRRC8D and 8E, also give rise to gain-of-function activity.
2. Instead of stating conditions C1 and C2, it's better to specify the key composition changes which give rise to hypotonicity by a simple cartoon illustration in the figure or stated in the figure legends. It's a bit confusing why C2 failed to induce VRAC currents in WT A+C group. From Figure 4G and 4H, it's also unclear if the mutants are still sensitive to hypotonicity without statistics between those groups.
3. For the basal activities of various mutants (LRRC8C or other subunits), it's important to examine their sensitivity to channel blockers (for example: dicumarol, DCPIB, or hypertonicity) to validate if they are indeed mediated by VRAC.
4. There are a lot of recent studies establishing important physiological function of VRAC particularly using mouse models. Acknowledging some of them does not in any way diminish the significance of the authors' current findings, with the respect to human physiology. So, the authors may want to put their findings in the appropriate context and modify: line 29; line 72; line 261-263. Indeed, the role of LRRC8A/SWELL1 in the brain and vasculature may link to the clinic features observed in these two patients.

We thank all three reviewers for their constructive comments, which we have considered in our revised manuscript.

Our revision contains the following new data:

- An experiment demonstrating that there is no detectable background due to non-specific binding of LRRC8A-GFP to streptactin resin displayed in Figure 4B.
- Patch-clamp experiments showing that endogenous and swelling-activated currents in channels containing either of the two LRRC8C disease variants share the same anion selectivity as WT. We also show that they are not permeable to larger anions such as glutamate. The data is displayed in Figure EV2F.
- Patch clamp experiments demonstrating the inhibition of endogenous currents of channels containing either disease variant by the VRAC inhibitor DCPIB, shown in Figure EV2G.
- Experiments characterizing the biochemical and functional properties of two truncated variants of LRRC8C that are found in the GnomAD database. Both variants were not associated with a disease phenotype. These constructs are somewhat shorter and longer than LRRC8C^{trunc}. The data is shown as novel Figure EV4. Whereas the shorter construct resulted in compromised expression, the longer protein folds and oligomerizes with similar properties as LRRC8C^{trunc}, when assayed by size exclusion chromatography (Fig EV4B). In both cases, electrophysiology experiments of respective constructs co-expressed with LRRC8A resulted in poor activation properties. This is in strong contrast to the pronounced gain of function obtained in equivalent channels containing the LRRC8C^{trunc} variant, providing a plausible explanation for the absent clinical phenotype.

We have also reformatted the manuscript according to the guidelines of the EMBO Journal. We have introduced a new Table summarizing the clinical phenotype as Table 1 and moved the cryo-EM table to the expanded view items as Table EV1. The three figures describing the cryo-EM data processing were moved to the appendix to reduce the number of expanded view figures. With Fig EV4 an additional figure was added to display data on the two novel LRRC8C constructs. In response to reviewer's requests, we have also included a discussion of the LRRC8A ébouriffé mutant and added several references on recent animal knock-out studies of LRRC8 genes that allowed insight into their involvement in physiological processes.

As detailed response to reviewer comments is provided below.

Referee #1:

In this very interesting paper Quinodoz et al. convincingly establish a causal link between de novo variants in LRRC8C and a clinical phenotype characterized by telangiectasia, microcephaly, metaphyseal dysplasia, eye abnormalities and short stature. This is a major step forward, as the authors are the first to identify mutations in LRRC8s in human pathology in the decade after the molecular identification of LRRC8 proteins as constituents of VRAC. The authors employ cryo-electron microscopy, biochemical techniques and patch-clamp electrophysiology, to show that the identified mutations introduce more flexibility to the heteromeric channel and hence mediate higher ion currents. The data appear of high quality and the writing is generally clear.

I have no major points.

Minor:

1. In order to influence channel function, LRRC8C

We have now performed the requested control experiment where we did not detect any background of LRRC8A-GFP in absence of LRRC8C-SBP constructs; we have added these results to the revised Figure 4B, C.

2. The "Impact of disease mutations on LRRC8A/C channel function" section does not read as self-contained: two different buffer combinations are mentioned, but the motivation to use both in parallel nor difference between them was explained. The manuscript would read more smoothly if it were stated that e.g. "in C1, the concentration of NaCl is reduced to lower the extracellular osmolarity, while to this end in C2, mannitol is omitted, while ion concentrations remain constant". It is also puzzling that in Sukalskaia et al, 2021 C2 caused activation of overexpressed LRRC8A/C channels in LRRC8-/- background, albeit the currents were smaller than those in WT HEK cells, but in the present manuscript, ICl_{vol} was seen only upon

overexpression of mutant channels in these ionic conditions. What can be the reason for such differences?

We have now renamed C1 into Δ chloride and C2 into Δ mannitol and described the main difference between isotonic and hypotonic buffers in the text.

Line 200-206:

‘In case of hLRRC8A/C^{WT}, this protocol resulted in a robust and reversible swelling-activated current response in one set of solutions with reduced salt concentration in hypotonic conditions (Δ salt), showing the characteristic properties of LRRC8A/C channels with pronounced outward rectification and weak inactivation at positive voltages (Voss *et al*, 2014) (Figs 4D and EV2A). No response of hLRRC8A/C^{WT} channels was detected in the second set of solutions with reduced mannitol concentration in hypotonic conditions (Δ mannitol) (Figs EV3C and D).’

The difference in the observed response of LRRC8A/C^{WT} in buffer system Δ mannitol (C2) is probably a consequence of the distinct orthologs used in the two studies. In the study described in *Sukalskaia et al, 2021*, we have used murine proteins, for which we found consistently stronger response upon overexpression in LRRC8^{-/-} cells than for their human counterparts used in this study.

3. Although the identified mutations are located far from the pore region, there are indications that strong VRAC activation may dampen its selectivity. Especially the truncating mutation is reminiscent of the result of proteolytic cleavage of the distantly related pannexin1 channels that allows for ATP transport (<https://doi.org/10.1038/s41586-020-2357-y>). It would be interesting to check whether the identified mutations affect halide permeability ratios or even facilitate permeation of organic substrates such as glutamate or taurine.

In our revised manuscript, we have included novel data showing a very similar selectivity as WT for the anions chloride and iodide and no detectable permeability for the larger anion glutamate in basal and activated currents of both mutants. This data is illustrated in Fig. EV2F.

We have described this finding in the results:

Line 209-211:

The basal currents share the same IV relationships and anion selectivity properties as currents of WT and mutated constructs after exposure to hypotonic conditions (Fig EV2A-F).

4. It would be nice to compare the effects of LRRC8C_{trunc} to a spontaneous mouse LRRC8A mutation *ébouriffé* that truncates the last 15 LRRs.

This mutation is interesting since, by reducing VRAC currents, LRRC8A^{ébouriffé} exerts the opposite phenotype of LRRC8C^{trunc}. However, it is still unclear whether this property is primarily the consequence of a low open probability or the inability of the protein to assemble and traffic to the plasma membrane. Remarkably, truncations of LRRC8C found in the gnomAD database, which result in protein of comparable length, do not exert a pathogenic phenotype. We have characterized one of these constructs and found a similar reduced activity as in LRRC8A^{ébouriffé} in LRRC8C^{t419} (see discussion below).

It should also be noted that the *ébouriffé* mouse genetic variant is recessive, i.e., mice are homozygous (Lalouette *et al*, 1996), whereas no abnormality was observed in heterozygous animals. The fact that heterozygous animals do not display a particular phenotype seems to indicate a hypomorphic effect of the variant. All this would fit with the observations of a number of individuals heterozygous for LRRC8C loss-of-function variants in the gnomAD database (see discussion further below). This further contrasts the dominant effect of LRRC8C^{trunc}.

We have introduced a comparison to the *ébouriffé* mutant in the discussion

Line 274-280:

In this respect, a previously described spontaneous mutation of LRRC8A leading to a premature stop removing the bulk of the LRRD in *ébouriffé* mice is relevant, since this construct, which is somewhat longer than the variant LRRC8C^{t419} leads to a similar severe reduction of activity (Fig EV4A, C). In this case, the low activity was assigned in part to the compromised assembly, which has interfered with the trafficking of LRRC8 channels to the plasma membrane (Luck *et al*, 2018; Platt *et al*, 2017). In contrast to LRRC8C^{trunc}, this variant is recessive, and no phenotype was observed in heterozygous mice.

5. The exact region where the mutations are located should be stated in the abstract.

We have mentioned that the mutations would be located at the boundary between the pore and a cytoplasmic domain in the abstract but are unfortunately limited in the detail of the description by the maximum word count allowed for this part of the manuscript.

6. In line 161 of the manuscript is a typing error. The authors refer to Figure EV3 which is actually EV4

We wanted to refer to both figures EV3 and EV4 and have corrected this in the manuscript.

Referee #2:

The manuscript describes two unrelated children with a de novo mutation in the LRRC8C channel, one with a missense (p.(Val390Leu)) and one frameshift mutation (p.(Leu400IlefsTer8)). These specific variants are not present in GnomAD and fall in a highly conserved region of the protein, which also is intolerant to variation in the normal population. The children show a remarkable similarity in phenotypes of blood vessels, brain, eye and bones. In addition, the two children share a similar in facial gestalt. LRRC8C has not before linked to a human syndrome. The authors propose to call this disorder LRRC8C-related TIMES Syndrome for telangiectasia, microcephaly, metaphyseal dysplasia, eye abnormalities and short stature. Expression studies of wt and mutated channels in HEK293 suggested proper folding and assembly of both mutants into the same monomeric heptameric state as WT. Cryo-EM studies of wt and mutants showed an increased flexibility of the membrane-inserted pore domain for both the mutants compared to the wildtype. Cellular function of LRRC channels is believed to work via heterodimers between LRRC8A and LRRC8C subunits. Next the authors convincingly show that both mutants could still be copurified as a heterodimer. Patch clamp experiments in LRRC8 knockout HEK293 cells transfected with mutants and wt LRRC8C showed an overactivity of the two mutant channels.

This manuscript describes the first human syndrome associated with mutations in LRRC8C, thus providing an important contribution to the field of human disease genetics. The description of LRRC8C mutations give insight in its function in the cell and during development. The functional studies show a mechanistic explanation of the gain-of-function of the mutated channels. These findings are of importance for both clinical genetics as well as provide important novel biological insights.

Major:

1. Two patients with de novo mutations in the same gene to define a novel syndrome is the absolute minimum, having (an) additional patient(s) would strengthen the manuscript. However, I see that the rarity of gain-of-function mutations might complicate this. The different types of mutation leading to the similar and specific phenotypes in both patients is a plus here.

We have attempted to identify additional patients sharing the same syndrome but did not succeed to find cases where the connection to the underlying mutations were similarly convincing. In light of the fact that these are spontaneous *de novo* mutations and the severity of the symptoms shorten the lifespan of the patients (one of the two patients is already deceased), we expect the disease to be extremely rare. The probability of identifying additional patients depends precisely on timely publication of our observation so that the gene can be included in gene panels that are sequenced for diagnostic purposes.

2. The patient mutations fall in a highly conserved region. The authors state that within this region the rate of missense changes is ~8-fold lower than the rest of the gene, and by the complete absence of frameshift or nonsense variants. I agree with this observation. The width of this conserved region, however, is narrow, as Gnomad lists truncating alleles in the normal population (e.g. around amino acid position 400: p.Trp340Ter, p.Glu357ArgfsTer10, p.Phe370CysfsTer11, p.Gln379Ter, p.Gln419Ter, p.Arg425Ter). In the current manuscript it is not clear how truncation at position 400 leads to so such diametrically different phenotypes compared to truncation at 379 and 419. These or similar variants in the population should be included in the functional experiments, as negative controls, to provide further evidence of the specific gain-of-function effects of the patient mutations.

3. Similar reasoning as comment 2 holds true for the missense variants present in the population, although additional functional experiments for the truncating variants would suffice in a revision.

In response to the request, we have investigated the properties of two mutations leading to the premature truncation of LRRC8C after residues 379 (LRRC8C^{t379}) and 419 (LRRC8C^{t419}). The associated data are included in the manuscript as novel Figure EV4 and are described in the results. The initial biochemical characterization of overexpressed homomeric constructs showed proper assembly of the longer LRRC8C^{t419} construct, as indicated by a peak at the appropriate elution volume of an oligomeric assembly, but not for the shorter construct LRRC8C^{t379}, which did not yield any protein after detergent extraction, indicating a problem with protein expression

or folding (Fig EV4B). We have then proceeded with functional recordings of LRRC8^{-/-} cells co-transfected with DNA coding for LRRC8A and either LRRC8C^{t379} or LRRC8C^{t419} by patch-clamp electrophysiology in buffer system Δsalt. In this system, we observed very low currents of both constructs in isotonic buffers that were not significantly above WT recorded in the same conditions (Fig EV4C). For these currents, we did not detect pronounced activation in response to swelling (Fig. EV4C), indicating a compromised structural behavior that is in stark contrast to the pronounced gain of function phenotype observed for LRRC8A/C^{trunc}. These findings offer a possible explanation for the apparent lack of clinical phenotype of individuals carrying these mutations in the gnomAD database.

In addition to Figure E7 we have included the following sentences:

Line 214-229:

Since the gnomAD database contains numerous nonsense variants of LRRC8C resulting in truncated proteins of different length that do not cause detectable symptoms, we have investigated the biochemical and functional properties of two variants that are closest to LRRC8C^{trunc}, one leading to a construct that is 21 residues shorter (termed LRRC8C^{t379}) and another that is 19 residues longer (termed LRRC8C^{t419}) (Fig EV4A). We have first investigated the biochemical properties of both constructs by monitoring their elution on size exclusion chromatography after overexpression and solubilization. In these experiments we found robust assembly similar to LRRC8C^{trunc} in case of the longer protein construct LRRC8C^{t419}, whereas no protein was detected in case of LRRC8C^{t379}, indicating a compromised expression and folding of the shorter protein construct (Fig EV4B). When investigating heteromeric channels containing either construct by patch-clamp electrophysiology in buffer system Δsalt, we found in both cases a consistent behavior with very low currents under isotonic conditions that are barely above WT and much smaller than in LRRC8C^{trunc} and little further activation upon change to hypotonic conditions, suggesting that both mutations resulted in severely compromised channels that do not share the strong gain of function exerted by the LRRC8^{trunc} or LRRC8^{V390L} variants (Fig EV4C).

Line 232-234:

The fact that this phenotype is not shared by constructs that are moderately smaller or larger than LRRC8C^{trunc}, points towards the importance of the mutated region as hotspot for channel activation.

Line 289-292:

The functional importance of the affected region as hotspot for activation is also reflected in the pronounced difference found in variants resulting from missense mutations located up- or downstream of the site mutated in LRRC8C^{trunc}, which in both cases leads to a pronounced loss of activity (Fig EV4C).

4. The discussion reads 'Haploinsufficiency, i.e. pathogenicity due to mutation-induced insufficient protein production, can be excluded as 220 control individuals carrying loss-of-function variants in exon 2 have been reported in the gnomAD database (v4.0) (Karczewski et al., 2020).' This contradicts the fact that GnomAD shows a significantly lower observed over expected ratio for both missense and nonsense variants in LRRC8C. Please discuss this in the manuscript.

We believe that the high pLI in gnomAD v.4.1 is artefactual and represents one of the many inconsistencies that are related to the recent expansion of the database, which now includes several poorly phenotyped cohorts. This is unfortunately a known fact and a current hot topic in the genetics community, as the reviewer probably knows already, to the point that most medical geneticists ignore v.4.1 and take as a reference v.2.1.1, which is the last version of gnomAD to include clinically ascertained control individuals.

LRRC8C is the perfect example of such problem: gnomAD v.2.1.1 reports for LRRC8C an observed/expect (o/e) ratio of 0.38 and a pLI (probability of being in the haploinsufficient class) of 0.00, making the gene an absolutely haplosufficient one. In contrast, gnomAD v.4.1 reports, for the same gene, an almost identical o/e ratio (0.35), but a pLI of 0.98, which is diametrically opposite to the previous value and puts LRRC8C in the class of the haploinsufficient genes. Obviously, this is either a computing artifact or the result of a massive inclusion of patients in v.4.1, which is unlikely. Most importantly, LRRC8C is a 3-exon gene and its coding sequence spans only exon 2 and exon 3. These exons have very different lengths, and exon 2 harbors only 5% of protein-coding sequence. Although we can be relatively sure that LoF variants in exon 2 would lead to nonsense-mediated mRNA decay (NMD) and therefore in no protein product, we cannot determine theoretically what would be the consequences on LRRC8C protein of LoFs in exon 3, since premature terminations in the last exon of a gene are generally thought to escape NMD and result in truncated proteins, which may be functional or not. We therefore do not know how pLI is calculated, given the particular architecture of this gene, for which ~95% of LoF DNA variants may or may not correspond to absence of protein. Notably, none of the LoF variants found in gnomAD are annotated with a low-confidence LoF tag which is normally associated with LoF variants in the last exon of genes.

For all these reasons, in our original text we discussed only LoF variants occurring in exon 2, noting that at least 220 control individuals would display no phenotype despite producing only 50% of the normal LRRC8C protein content. In light of the recent findings related to patients

being included in v.4.1 of gnomAD, the possibility of computational mistakes related to this same version (see: <https://discuss.gnomad.broadinstitute.org/c/general/4>), and to avoid entering the ongoing quarrel in the field of medical genetics about whether or not to use v.4.1, we have reassessed our findings by taking into account data from gnomAD v2.1.1 (which reports variants related to control individuals only) and modified the text in the Discussion as follows:

Line 249-252:

Haploinsufficiency, i.e. pathogenicity due to variant-induced insufficient protein production, can be excluded as two control individuals carrying loss-of-function variants in exon 2 (the first coding exon) have been reported in the gnomAD database (v.2.1.1) (Karczewski et al., 2020).

Minor:

1. Please refrain from using the term mutation (abstract). The term "sequence variation" is used to prevent confusion with the term "mutation" and "polymorphism", mutation meaning "change" in some disciplines and "disease-causing change" in others and polymorphism meaning "non disease-causing change" or "change found at a frequency of 1% or higher in the population".

We have replaced the term mutation with sequence variations in the abstract and throughout the text but occasionally use the term mutation when talking about expressed protein constructs.

2. Please provide genomic mutation information of the two mutations (hg38)

We have added the information in the text:

Line 88-92:

This DNA variant, NM_032270.4:c.1197dup, p.(Leu400IlefsTer8) (NC_000001.11:g.89713767dup [hg38], NC_000001.10:g.90179326dup [hg19]), produced a shift of the reading frame and thus the change of Leu 400 to Ile, the insertion of six amino acids followed by a premature termination codon (Fig 2A).

Line 100-103:

The female patient carried a different monoallelic variant in LRRC8C, NM_032270.4:c.1168G>C, p.(Val390Leu) (NC_000001.11:g.89713738G>C [hg38],

NC_000001.10:g.90179297G>C [hg19]), or LRRC8CV390L, which also occurred in a de novo manner (Figs 1A,C and E).

3. Clinical geneticists would be helped with a (supplemental) table summarizing the clinical features for each patient

We have included such table in our revision as Table 1.

4. Line 989: typo: Iso

We have corrected the typo.

5. Line 998: typo: bléomycin

We have corrected the typo.

Referee #3:

In this manuscript, Mathieu Quinodoz and colleagues identified two de novo mutations in LRRC8C gene in two similar cases of a congenital syndrome affecting multiple systems. Based on structural and electrophysiological analysis on the two mutants, the authors propose that the mutations impair VRAC channel gating and result in gain-of-function, which likely underlying the disease pathogenesis. The findings are novel and significant because it represents the first clear evidence linking VRAC/LRRC8 channel mutations to human pathology. It also suggests the linker region between the pore domain and LRR domains as a critical element in VRAC channel gating.

Here are a few major comments:

1. Because the valine to leucine mutation is very subtle, it would add a lot of confidence to the authors' conclusion if similar mutations in LRRC8A and other LRRC8 genes, particular LRRC8D and 8E, also give rise to gain-of-function activity.

We are currently engaged in a collaborative study addressing the role of the same protein region in LRRC8A. Our collaborators have found a very similar strong activating effect of mutations in

the same position in this obligatory subunit, supporting our findings described in this manuscript. We are currently also investigating the activation properties of equivalent mutations in LRRC8D and E as part of an independent study. Preliminary results on the D constructs are at this stage less conclusive. As these experiments are part of ongoing studies, we prefer not to include them in our manuscript as we think that they require a more systematic in-depth investigation.

2. Instead of stating conditions C1 and C2, it's better to specify the key composition changes which give rise to hypotonicity by a simple cartoon illustration in the figure or stated in the figure legends. It's a bit confusing why C2 failed to induce VRAC currents in WT A+C group. From Figure 4G and 4H, it's also unclear if the mutants are still sensitive to hypotonicity without statistics between those groups.

We have renamed C1 into Δ salt and C2 into Δ mannitol to account for the major changes between isotonic and hypotonic conditions. As described above, we were also surprised that the Δ mannitol conditions did not evoke a detectable response for LRRC8A/C^{WT} transfection on the human proteins, in contrast to previous studies on mouse orthologs, where we have observed stronger response in the same buffer system in response to changes to hypotonic conditions.

After analysis, we do detect statistically significant activation for both constructs in Δ mannitol conditions and LRRC8A/C^{V290L} but not LRRC8A/C^{trunc} upon shift to isotonic solutions in Δ chloride conditions, probably due to the larger basal activity of the latter construct. However, we do not want to conclude that any of the two disease constructs would be insensitive to hypotonicity in a physiological context.

3. For the basal activities of various mutants (LRRC8C or other subunits), it's important to examine their sensitivity to channel blockers (for example: dicumarol, DCPIB, or hypertonicity) to validate if they are indeed mediated by VRAC.

We have now investigated the effect of addition of pore blocker DCPIB under isotonic conditions and found strong inhibition in both cases. The data are displayed in Figure EV5G.

We have also changed the text to:

Line 209-213:

The basal currents share the same IV relationships and anion selectivity properties as currents of WT and mutated constructs obtained after exposure to hypotonic conditions (Fig EV2A-F). Additionally, their inhibition with the pore blocker DCPIB provides further evidence that they originate from open VRAC channels (Fig EV2G).

4. There are a lot of recent studies establishing important physiological function of VRAC particularly using mouse models. Acknowledging some of them does not in any way diminish the significance of the authors' current findings, with the respect to human physiology. So, the authors may want to put their findings in the appropriate context and modify: line 29; line 72; line 261-263. Indeed, the role of LRRC8A/SWELL1 in the brain and vasculature may link to the clinic features observed in these two patients.

We have rephrased this part in the discussion and included several references in our revision.

Line 297-302:

The ubiquitous expression of VRACs in vertebrate cells has suggested critical roles for the channel, with several studies using knock out animals providing initial insight into specific functions in which VRACs may participate (Alghanem *et al*, 2021; Balkaya *et al*, 2023; Chu *et al*, 2023; Concepcion *et al*, 2022; Knecht *et al*, 2024; Lopez-Cayuqueo *et al*, 2022). However, despite these extensive studies, many physiologic processes it is involved in have remained enigmatic.

References

- Alghanem AF, Abello J, Maurer JM, Kumar A, Ta CM, Gunasekar SK, Fatima U, Kang C, Xie L, Adeola O *et al* (2021) The SWELL1-LRRC8 complex regulates endothelial AKT-eNOS signaling and vascular function. *Elife* 10
- Balkaya M, Dohare P, Chen S, Schober AL, Fidaleo AM, Nalwalk JW, Sah R, Mongin AA (2023) Conditional deletion of LRRC8A in the brain reduces stroke damage independently of swelling-activated glutamate release. *iScience* 26: 106669
- Chu J, Yang J, Zhou Y, Chen J, Chen KH, Zhang C, Cheng HY, Koylass N, Liu JO, Guan Y *et al* (2023) ATP-releasing SWELL1 channel in spinal microglia contributes to neuropathic pain. *Sci Adv* 9: eade9931
- Concepcion AR, Wagner LE, 2nd, Zhu J, Tao AY, Yang J, Khodadadi-Jamayran A, Wang YH, Liu M, Rose RE, Jones DR *et al* (2022) The volume-regulated anion channel LRRC8C suppresses T cell function by regulating cyclic dinucleotide transport and STING-p53 signaling. *Nat Immunol* 23: 287-302
- Knecht DA, Zeziulia M, Bhavsar MB, Puchkov D, Maier H, Jentsch TJ (2024) LRRC8/VRAC volume-regulated anion channels are crucial for hearing. *J Biol Chem* 300: 107436
- Lalouette A, Lablack A, Guenet JL, Montagutelli X, Segretain D (1996) Male sterility caused by sperm cell-specific structural abnormalities in ebouriffe, a new mutation of the house mouse. *Biol Reprod* 55: 355-363
- Lopez-Cayuqueo KI, Planells-Cases R, Pietzke M, Oliveras A, Kempa S, Bachmann S, Jentsch TJ (2022) Renal Deletion of LRRC8/VRAC Channels Induces Proximal Tubulopathy. *J Am Soc Nephrol* 33: 1528-1545
- Luck JC, Puchkov D, Ullrich F, Jentsch TJ (2018) LRRC8/VRAC anion channels are required for late stages of spermatid development in mice. *J Biol Chem* 293: 11796-11808
- Platt CD, Chou J, Houlihan P, Badran YR, Kumar L, Bainter W, Poliani PL, Perez CJ, Dent SYR, Clapham DE *et al* (2017) Leucine-rich repeat containing 8A (LRRC8A)-dependent volume-regulated anion channel activity is dispensable for T-cell development and function. *J Allergy Clin Immunol* 140: 1651-1659 e1651
- Voss FK, Ullrich F, Munch J, Lazarow K, Lutter D, Mah N, Andrade-Navarro MA, von Kries JP, Stauber T, Jentsch TJ (2014) Identification of LRRC8 heteromers as an essential component of the volume-regulated anion channel VRAC. *Science* 344: 634-638

Dear Prof. Dutzler,

I am pleased to inform you that your manuscript has been accepted for publication in the EMBO Journal. Thank you for trusting us with this work.

Yours sincerely,

William Teale, PhD
Editor
The EMBO Journal
w.teale@embojournal.org

P.s. The review reports of the revised version of the manuscript are copied below for your records.

Referee #1:

The authors addressed all our comments and I believe that this interesting paper is now ready for publication.

Referee #2:

My concerns have been addressed in this revised version of the manuscript in an elegant manner.

Referee #3:

The authors addressed all of the questions. Here are a few minor comments/suggestions:

1. RE both reviewer #1 and #3's comments, the authors may consider to add a simple note as stated in the rebuttal at line 210 commenting on the surprising finding that no A/C WT currents were observed under hypotonic condition (Δ mannitol). This will help avoid confusion and may also provide useful information (mouse vs human) to the field.
2. Line 216: the evidence supporting DCPIB as a pore blocker is very strong, and it is generally not referred as such. The authors may consider rephrase this sentence.
3. Figure 4H: some labels on the x-axis are missing.
4. Figure EV2G: some labels are missing?
